# FASTERVIT: FAST VISION TRANSFORMERS WITH HIERARCHICAL ATTENTION

**Ali Hatamizadeh, Greg Heinrich, Hongxu Yin, Andrew Tao, Jose M. Alvarez, Jan Kautz, Pavlo Molchanov**
NVIDIA
{ahatamizadeh, pmolchanov}@nvidia.com

## ABSTRACT

We design a new family of hybrid CNN-ViT neural networks, named FasterViT, with a focus on high image throughput for computer vision (CV) applications. FasterViT combines the benefits of fast local representation learning in CNNs and global modeling properties in ViT. Our newly introduced Hierarchical Attention (HAT) approach decomposes global self-attention with quadratic complexity into a multi-level attention with reduced computational costs. We benefit from efficient window-based self-attention. Each window has access to dedicated carrier tokens that participate in local and global representation learning. At a high level, global self-attentions enable the efficient cross-window communication at lower costs. FasterViT achieves a SOTA Pareto-front in terms of accuracy and image throughput. We have extensively validated its effectiveness on various CV tasks including classification, object detection and segmentation. We also show that HAT can be used as a plug-and-play module for existing networks and enhance them. We further demonstrate significantly faster and more accurate performance than competitive counterparts for images with high resolution.

Code is available at https://github.com/NVlabs/FasterViT.

## 1 INTRODUCTION

Vision Transformers (ViTs) (Dosovitskiy et al., 2020) have recently become popular in computer vision and achieved superior performance in various applications such as image classification (Liu et al., 2021; Dong et al., 2022; Lin et al., 2017), object detection (Zhang et al., 2021b; Fang et al., 2021) and semantic segmentation (Xie et al., 2021; Cheng et al., 2021). In addition to learning more uniform local and global representations across their architecture when compared to Convolutional Neural Networks (CNNs), ViTs scale properly to large-scale data and model sizes (Raghu et al., 2021; Paul & Chen, 2022). Recently, several efforts (He et al., 2022; Xie et al., 2022) have also shown the exceptional capability of ViTs in self-supervised learning of surrogate tasks such as masked image modeling which may significantly enhance the performance of downstream applications. Despite these advantages, lack of inductive bias in pure ViT models may require more training data and impede performance (Xu et al., 2021b). Hybrid architectures, which consist of both CNN and ViT-based components, could address this problem and achieve competitive performance without needing large-scale training datasets (Dosovitskiy et al., 2020) or other techniques such as knowledge distillation (Touvron

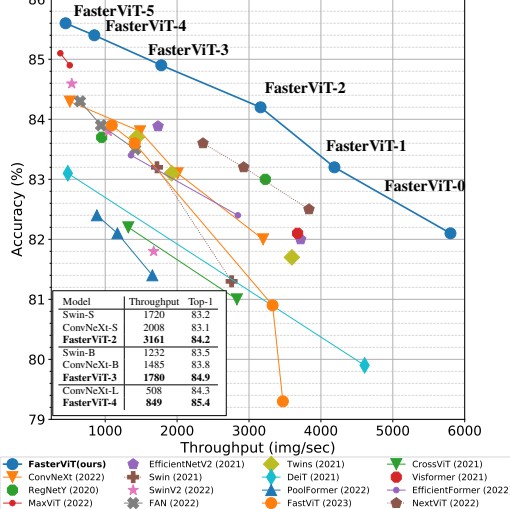

| Model | Throughput | Top-1 |
|---|---|---|
| Swin-S | 1720 | 83.2 |
| ConvNeXt-S | 2008 | 83.1 |
| **FasterViT-2** | **3161** | **84.2** |
| Swin-B | 1232 | 83.5 |
| ConvNeXt-B | 1485 | 83.8 |
| **FasterViT-3** | **1780** | **84.9** |
| ConvNeXt-L | 508 | 84.3 |
| **FasterViT-4** | **849** | **85.4** |

Figure 1: Comparison of image throughput and ImageNet-1K Top-1 accuracy. Throughput is measured on A100 GPU with batch size of 128.

et al., 2021a). An integral component of ViTs is the self-attention mechanism (Vaswani et al., 2017; Dosovitskiy et al., 2020) which enables modeling of both short and long-range spatial dependencies. However, the quadratic computational complexity of self-attention significantly impacts the efficiency and hinders its use for applications with high-resolution images. In addition, contrary to the isotropic architecture (*i.e.*, same feature resolution with no downsampling) of the original ViT model, learning feature representations in a multi-scale manner typically yields better performance (Fan et al., 2021; Wang et al., 2022), specifically for downstream applications (*e.g.*, detection, segmentation).

To address these issues, Swin Transformer (Liu et al., 2021) proposed a multi-scale architecture in which self-attention is computed in local windows, and window-shifting allows for interaction of different regions. However, due to the limited receptive field of these local regions and small area of coverage in window shifting (Liu et al., 2021; Lin et al., 2017), cap-turing cross-window interactions and model-ing the long-range spatial dependencies become challenging for large-resolution input features. Furthermore, using self-attention blocks in early stages with larger resolution may impact the im-age throughput due to the increased number of local windows. Recently, the Swin Transformer V2 model (Liu et al., 2022a) was proposed to address training instabilities on high-resolution images by improving the self-attention mecha-nism. However, in addition to having a lower

Figure 2: Visualization of the proposed Hierarchical Attention in the feature space. By performing local window attention and hierarchical attention we can achieve global information propagation at reduced costs.

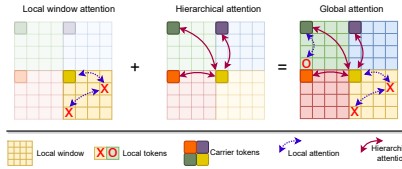

image throughput compared to the Swin Transformer (Liu et al., 2021), Swin Transformer V2 still relies on the original window-shifting mechanism for cross-interaction of different windows, which becomes less effective with large image sizes.

In this work, we attempt to address these issues and propose a novel hybrid architecture, denoted as FasterViT, which is tailored for high-resolution input images, while maintaining a fast image throughput. FasterViT consists of four different stages in which the input image resolution is reduced by using a strided convolutional layer, while doubling the number of feature maps. We propose to leverage residual convolutional blocks in the high-resolution stages of the architecture (*i.e.*, stage 1, 2), while employing transformer-blocks in later stages (*i.e.*, stage 3, 4). This strategy allows for fast generation of high-level tokens which can be further processed with the transformer-based blocks. For each transformer block, we use an interleaved pattern of local and, newly proposed, Hierarchical Attention blocks to capture both short and long-range spatial dependencies and efficiently model the cross-window interactions. Specifically, our proposed Hierarchical Attention (see Fig. 2) learns carrier tokens as a summary of each local window and efficiently models the cross-interaction between these regions. The computational complexity of the Hierarchical Attention grows almost linearly with input image resolution, as the number of regions increases, due to the local windowed attention being the compute bottleneck. Hence, it is an efficient, yet effective way of capturing long-range information with large input features.

We have extensively validated the effectiveness of the proposed FasterViT model on various image tasks and datasets such as ImageNet-1k for image classification, MS COCO for object detection and instance segmentation and ADE20K dataset for semantic segmentation. FasterViT achieves state-of-the-art performance considering the trade-off between performance (*e.g.*, ImageNet-1K top-1 accuracy) and image throughput (see Fig. 1). To demonstrate the scalability of FasterViT for larger datasets, we have also pre-trained FasterViT on ImageNet-21K dataset and achieved state-of-the-art performance when fine-tuning and evaluating on larger-scale resolutions.

The summary of our contributions is as follows:

- We introduce FasterViT, which is a novel hybrid vision transformer architecture designed for an optimal trade-off between performance and image throughput. FasterViT scales effectively to higher resolution input images for different dataset and model sizes.
- We propose the Hierarchical Attention module which efficiently captures the cross-window interactions of local regions and models the long-range spatial dependencies.
- FasterViT achieves a new SOTA Pareto front in terms of image throughput and accuracy trade-off and is significantly faster than comparable ViT-based architectures yielding signifi-

cant speed-up compared to recent SOTA models. It also achieves competitive performance on downstream tasks of detection and instance segmentation on MS COCO dataset and semantic segmentation on ADE20K dataset.

## 2 RELATED WORK

**Vision Transformers.** Oriented from the language processing domain, the first application of transformer architecture to vision task immediately offers an inspiring demonstration of the high efficacy of attention across image patches across varying scenarios (Dosovitskiy et al., 2020). The appealing strength of vision transformer and its architecture and logic simplicity has therefore triggered a quickly evolving literature in the past two years, where ViT performance is quickly boosted by an erupting new set of innovations: network-wise leveraging knowledge distillation for data-efficient training as in DeiT (Touvron et al., 2021a), hybriding convolution and self-attention for enhanced inductive biases as in LeViT (Graham et al., 2021), imposing CNN-inspired pyramid rules on ViTs (Wang et al., 2021; 2022), along with component-wise improvements such as improved token utilization as in T2T-ViT (Yuan et al., 2021), enhanced positional embedding (Chu et al., 2023), local window attention as shown in the inspiring work of the Swin family (Liu et al., 2021; 2022a) and CSwin (Dong et al., 2022), global attention in GCViT (Hatamizadeh et al., 2023), among many other architectural insights (Chu et al., 2021a; Zhang et al., 2021a; Yuan et al., 2022). Along with the increasing capacity comes the increasing computation burden. As similarly facing challenges in scaling up the models in language tasks (e.g., from BERT-Large 0.3B (Devlin et al., 2019), to Megatron-LM 8.3B (Shoeybi et al., 2019), and Switch-Transformer1.6T (Fedus et al., 2022)), scaling up vision transformers is also a highly challenging but highly important task (Dai et al., 2021; Liu et al., 2022a) due to the attention-extensive nature of transformers, urging efficiency for pervasive usage.

**Towards Enhanced Efficiency.** Boosting up ViT efficiency has therefore been a very vibrant area. One stream of approach roots in the efficient deep learning literature that cuts down on network complexity leveraging popular methods such as efficient attention (Bolya et al., 2022; Lu et al., 2021; Cai et al., 2022), network compression (Chen et al., 2021b;c; Liang et al., 2022; Yang et al., 2021a), dynamic inference (Yin et al., 2022; Rao et al., 2021), operator adaptation (Molchanov et al., 2022), token merging and manipulations (Marin et al., 2021; Xu et al., 2022), etc. These methods can yield off-the-shelf speedups on target ViT backbones, but are also limited to the original backbone's accuracy and capacity. Another stream of work, on the other hand, focuses on designing new ViT architectures with enhanced efficiency as an original design objective. For example, EfficientFormer (Li et al., 2022) entails mobile applications through dimension-consistent re-design of transformer block and removing redundant architectural components. VisFormer (Chen et al., 2021d) transits computation extensive transformer to a convolutional counterpart for enhanced vision efficiency. CrossViT (Chen et al., 2021a) learns multi-scale features and utilizes small/large-patch backed tokens that are channeled by efficient attention, offering linear time and memory complexity. Even with such a rapid progress in literature, enabling efficient ViTs remains a significant challenge, where we next further push the Pareto front of faster ViT on top of prior art by a large margin. Note that we focus on the second stream of architectural redesign for efficiency boost, and consider a joint exploration with the first acceleration stream of method like compression as orthogonal and fruitful future work.

**Global Self-Attention.** A number of efforts have introduced global self-attention to capture more contextual information. In NLP (*i.e., 1D*), BigBird (Zaheer et al., 2020) and LongFormer (Beltagy et al., 2020) proposed to select special tokens (*i.e. non-learnable*) as global tokens to attend to other tokens via a sliding-window dense self-attention. In computer vision, EdgeViT (Pan et al., 2022), Twins (Chu et al., 2021a) and Focal Transformer (Yang et al., 2021b) proposed hierarchical-like attention mechanisms which rely on heuristic token aggregation in the forms of pooling (Yang et al., 2021b) or linear projection (Pan et al., 2022; Chu et al., 2021a). There are three key differences between these efforts and our proposed hierarchical attention: (1) as opposed to using a pre-defined mechanism to select the global tokens (*e.g., random*), we propose to learn these tokens (*i.e., carrier token*) via summarizing the role of each region in the input feature space (2) we propose learnable token aggregation and propagation mechanisms by computing self-attention among carrier tokens (3) as opposed to using dense/dilated self-attention, our proposed HAT uses local window-based self-attention and has a smaller computational complexity.

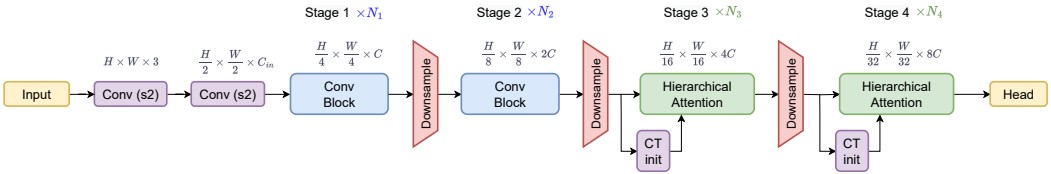

Figure 3: Overview of the FasterViT architecture. We use a multi-scale architecture with CNN and transformer-based blocks in stages 1, 2 and 3, 4, respectively. Best viewed in color.

## 3 FASTERVIT

### 3.1 DESIGN PRINCIPALS

We next detail our FasterViT architecture, offering Pareto accuracy-latency trade-off. We focus on highest throughput for computer vision tasks on mainstream off-the-shelf hardware such as GPUs that excel in parallel computing. Computation in this case involves a set of streaming multiprocessors (SMs) with CUDA and Tensor cores as computation units. It requires frequent data transfer for calculation and can be impacted by data movement bandwidth. As such, operations bounded by computation are math-limited, while those bounded by memory transfer are memory-limited. It requires a careful balance between the two to maximize throughput.

In hierarchical vision models, spatial dimension of intermediate representation shrinks as inference proceeds. Initial network layers mostly have larger spatial dimensions and fewer channel (*e.g.*, $112 \times 112 \times 64$), making them memory-bound. This makes a better fit for compute-intensive operations, such as dense convolution instead of depth-wise/sparse counterparts that impose extra transfer cost. Also operations not representable in matrix manipulation forms, *e.g.*, non-linearity, pooling, batch normalization, are also memory-bound and shall be minimized for usage. On the contrary, later layers tend to be math-limited with computationally expensive operations. For example, hierarchical CNNs have feature maps of size $14 \times 14$ with high dimensional kernels. This leaves room for more expressive operations such as Layer Normalization, squeeze-and-excitation, or attention, with fairly small effect on throughput. Guided by these insights we design a novel architecture that will benefit all stages from accelerated computing hardware.

### 3.2 ARCHITECTURE

Our overall design is shown in Fig. 3. It exploits convolutional layers in the earlier stages that operate on higher resolution. The second half of the model relies on novel hierarchical attention layers to reason spatially across the entire feature maps. In this design, we optimize the architecture for compute and throughput. As a result, the first half of the network and downsampling blocks make use of dense convolutional kernels. We also avoid squeeze-and-excitation operators and minimize Layer Normalization for higher resolution stages (*i.e.*, 1, 2), as these layers tend to be math-limited. Later stages (*i.e.*, 3, 4) in the architecture tend to be math-limited as GPU hardware spends more time on compute compared to the memory transfer cost. As a result, applying multi-head attention will not be a bottleneck.

### 3.3 FASTERVIT COMPONENTS

**Stem** An input image $\mathbf{x} \in \mathbb{R}^{H \times W \times 3}$ is converted into overlapping patches by two consecutive $3 \times 3$ convolutional layers, each with a stride of 2, which project them into a $D$-dimensional embedding. The embedded tokens are further batch-normalized (Ioffe & Szegedy, 2015) and use the ReLU activation function after each convolution.

**Downsampler Blocks** FasterViT follows the hierarchical structure: the spatial resolution is reduced by 2 between stages by a downsampling block. We apply 2D layer normalization on spatial features, followed by a convolutional layer with a kernel of $3 \times 3$ and a stride of two.

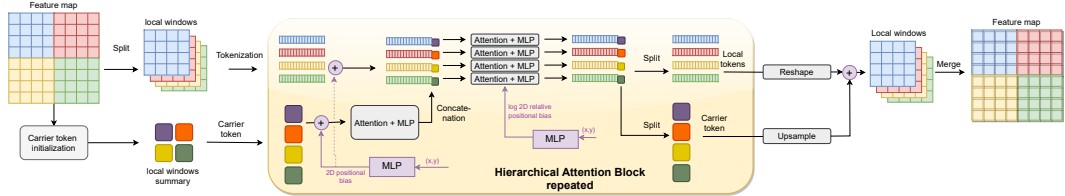

Figure 4: Proposed Hierarchical Attention block. Carrier tokens (CT) learn a summary of each local window and facilitate global information exchange between local windows. Local window tokens only have access to a dedicated subset of CT for efficient attention. CT undergo full self-attention to enable cross-window attention. "Attention" stands for MHSA (Vaswani et al., 2017), MLP for multi-layer perceptron. Best viewed in color.

**Conv Blocks**   Stage 1 and 2 consist of residual convolutional blocks, which are defined as

$$\hat{\mathbf{x}} = \text{GELU}(\text{BN}(\text{Conv}_{3\times3}(\mathbf{x}))),$$
$$\mathbf{x} = \text{BN}(\text{Conv}_{3\times3}(\hat{\mathbf{x}})) + \mathbf{x}, \tag{1}$$

where BN denotes batch normalization (Ioffe & Szegedy, 2015). Following the design principles, these convolutions are dense.

**Hierarchical Attention**   In this work, we propose a novel formulation of windowed attention, summarized in Fig 2 and detailed presentation in Fig 4. We start with local windows introduced in Swin Transformer (Liu et al., 2021). Then, we introduce a notion of *carrier tokens* (CTs) that play the summarizing role of the entire local window. The first attention block is applied on CTs to summarize and pass global information. Then, local window tokens and CTs are concatenated, such that every local window has access only to its own set of CTs. By performing self attention on concatenated tokens we facilitate local and global information exchange at reduced cost. By alternating sub-global (CTs) and local (windowed) self-attention we formulate a concept of *hierarchical attention*. Conceptually, CTs can be further grouped into windows and have a higher order of carrier tokens.

Assume we are given an input feature map $\mathbf{x} \in \mathbb{R}^{H \times W \times d}$ in which $H$, $W$ and $d$ denote the height, width and number of feature maps, let us set $H = W$ for simplicity. We first partition the input feature map into $n \times n$ local windows with $n = \frac{H^2}{k^2}$, where $k$ is the window size, as:

$$\hat{\mathbf{x}}_{\mathbf{l}} = \text{Split}_{k \times k}(\mathbf{x}). \tag{2}$$

The key idea of our approach is the formulation of *carrier tokens* (CTs) that help to have an attention footprint much larger than a local window at low cost. At first, we initialize CTs by pooling to $L = 2^c$ tokens per window:

$$\hat{\mathbf{x}}_{\mathbf{c}} = \text{Conv}_{3\times3}(\mathbf{x}),$$
$$\hat{\mathbf{x}}_{\mathbf{ct}} = \text{AvgPool}_{H^2 \to n^2 L}(\hat{\mathbf{x}}_{\mathbf{c}}), \tag{3}$$

where $\text{Conv}_{3\times3}$ represents efficient positional encoding inspired by (Chu et al., 2021b) and used in Twins (Chu et al., 2021a). $\hat{\mathbf{x}}_{\mathbf{ct}}$ and AvgPool denote the carrier tokens and feature pooling operation, respectively; $c$ is set to 1, but can be changed to control latency. The current approach with conv+pooling gives flexibility with the image size. These pooled tokens represent a summary of their respective local windows, we set $L << k$. The procedure of CT initialization is performed only once for every resolution stage. Note that every local window $\hat{\mathbf{x}}_{\mathbf{l}}$ has unique set of carrier tokens, $\hat{\mathbf{x}}_{\mathbf{ct,l}}$, such that $\hat{\mathbf{x}}_{\mathbf{ct}} = \{\hat{\mathbf{x}}_{\mathbf{ct,l}}\}_{l=0}^{n}$.

In every HAT block, CTs undergo the attention procedure:

$$\hat{\mathbf{x}}_{\mathbf{ct}} = \hat{\mathbf{x}}_{\mathbf{ct}} + \gamma_1 \cdot \text{MHSA}(\text{LN}(\hat{\mathbf{x}}_{\mathbf{ct}})),$$
$$\hat{\mathbf{x}}_{\mathbf{ct}} = \hat{\mathbf{x}}_{\mathbf{ct}} + \gamma_2 \cdot \text{MLP}_{d \to 4d \to d}(\text{LN}(\hat{\mathbf{x}}_{\mathbf{ct}})), \tag{4}$$

where LN represents layer normalization (Ba et al., 2016), MHSA represents multi-head self attention (Vaswani et al., 2017), $\gamma$ is a learnable per-channel scale multiplier (Touvron et al., 2021b), $\text{MLP}_{d \to 4d \to d}$ is a 2-layer MLP with GeLU (Hendrycks & Gimpel, 2016) activation function.

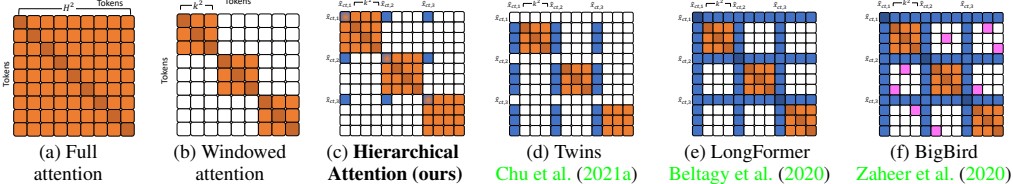

Figure 5: Attention map comparison for a feature map of size $H \times H \times d$. ☐ - no attention, ■ - normal token attention, ■ - carrier token attention, ■ - random token attention. Full attention (a) has complexity of $O(H^4 d)$, windowed attention significantly reduces it to $O(k^2 H^2 d)$ but lacks global context.

Next, in order to model short-long-range spatial information, we compute the interaction between the local and carrier tokens, $\hat{\mathbf{x}}_{\mathbf{l}}$ and $\hat{\mathbf{x}}_{\mathbf{ct,l}}$, respectively. At first, local features and CTs are concatenated. Each local window only has access to its corresponding CTs:

$$\hat{\mathbf{x}}_{\mathbf{w}} = \text{Concat}(\hat{\mathbf{x}}_{\mathbf{l}}, \hat{\mathbf{x}}_{\mathbf{ct,l}}). \tag{5}$$

These tokens undergo another set of attention procedure:

$$\begin{aligned} \hat{\mathbf{x}}_{\mathbf{w}} &= \hat{\mathbf{x}}_{\mathbf{w}} + \gamma_1 \cdot \text{MHSA}(\text{LN}(\hat{\mathbf{x}}_{\mathbf{w}})), \\ \hat{\mathbf{x}}_{\mathbf{w}} &= \hat{\mathbf{x}}_{\mathbf{w}} + \gamma_2 \cdot \text{MLP}_{d \to 4d \to d}(\text{LN}(\hat{\mathbf{x}}_{\mathbf{w}})). \end{aligned} \tag{6}$$

Finally, tokens are further split back and used in the subsequent hierarchical attention layers:

$$\hat{\mathbf{x}}_{\mathbf{l}}, \hat{\mathbf{x}}_{\mathbf{ct,l}} = \text{Split}(\hat{\mathbf{x}}_{\mathbf{w}}), \tag{7}$$

Procedures described in Equations 4-7 are iteratively applied for a number of layers in the stage. To further facilitate long-shot-range interaction, we perform *global information propagation*, similar to the one in (Pan et al., 2022) in the end of the stage. Finally, the output of the stage is computed as:

$$\mathbf{x} = \text{Upsample}_{n^2 L \to H^2}(\hat{\mathbf{x}}_{\mathbf{ct,l}}) + \text{Merge}_{n^2 k^2 \to H^2}(\hat{\mathbf{x}}_{\mathbf{l}}) \tag{8}$$

MHSAs performed in Eq. 4 and 6 are token position invariant, however, the location of features in the spatial dimension are clearly informative. To address this, we first add absolute positional bias directly to CTs and local window tokens. We are inspired by SwinV2 (Liu et al., 2022a) and employ a 2-layer MLP to embed absolute 2D token location into feature dimension. Then, to facilitate image-like locality inductive bias we enhance the attention with log space relative positional bias from SwinV2 (Liu et al., 2022a) (2-layer MLP). It ensures that the relative position of tokens contribute to shared attention patterns. This approach yields flexibility regarding image size, as the positional encoding is interpolated by the MLP, and hence a trained model can be applied to any input resolution.

An attention map comparison between efficient global-local self attention is shown in Fig. 5. The proposed hierarchical attention splits full attention into local and sub-global, both compressible to 2 dense attentions. Carrier tokens participate in both attentions and facilitate information exchange.

**Complexity Analysis of HAT** The key features of the efficiency of our approach are (i) separation of attentions and (ii) local windows only have access to their CTs. The complexity of the most conventional and popular full attention is $O(H^4 d)$. Partitioning the feature size into windows of size $k$, and running the attention, simplifies the attention to $O(k^2 H^2 d)$ as proposed in (Liu et al., 2021). It is well known that such windowed attention is more efficient but lacks global feature interaction. Our approach takes this one step further and is based on carrier tokens that summarize and interact over the entire feature map, to remedy for missing global communication. Given $L$ total carrier tokens per window, local window complexity is $O((k^2 + L)H^2 d)$. Local (windowed) attention is followed by attention on carrier tokens with complexity $O((\frac{H^2}{k^2}L)^2 d)$. The total cost of both attentions is $O(k^2 H^2 d + LH^2 d + \frac{H^4}{k^4}L^2 d)$.

An orthogonal approach for multilevel attention is to provide access to subsampled global information inside local attention. For example, Twins (Chu et al., 2021a) subsamples global feature map and

Table 1: Comparison of classification benchmarks on **ImageNet-1K** dataset (Deng et al., 2009). Image throughput is measured on A100 GPUs with batch size of 128.

| Model | Image Size (Px) | #Param (M) | FLOPs (G) | Throughput (Img/Sec) | Top-1 (%) |
|---|---|---|---|---|---|
| Conv-Based | | | | | |
| ConvNeXt-T Liu et al. (2022b) | 224 | 28.6 | 4.5 | 3196 | 82.0 |
| ConvNeXt-S Liu et al. (2022b) | 224 | 50.2 | 8.7 | 2008 | 83.1 |
| ConvNeXt-B Liu et al. (2022b) | 224 | 88.6 | 15.4 | 1485 | 83.8 |
| RegNetY-040 Radosavovic et al. (2020) | 288 | 20.6 | 6.6 | 3227 | 83.0 |
| ResNetV2-101 Wightman et al. (2021) | 224 | 44.5 | 7.8 | 4019 | 82.0 |
| EfficientNetV2-S Tan & Le (2021) | 384 | 21.5 | 8.0 | 1735 | 83.9 |
| Transformer-Based | | | | | |
| Swin-T Liu et al. (2021) | 224 | 28.3 | 4.4 | 2758 | 81.3 |
| Swin-S Liu et al. (2021) | 224 | 49.6 | 8.5 | 1720 | 83.2 |
| SwinV2-T Liu et al. (2022a) | 256 | 28.3 | 4.4 | 1674 | 81.8 |
| SwinV2-S Liu et al. (2022a) | 256 | 49.7 | 8.5 | 1043 | 83.8 |
| SwinV2-B Liu et al. (2022a) | 256 | 87.9 | 15.1 | 535 | 84.6 |
| Twins-B Chu et al. (2021a) | 224 | 56.1 | 8.3 | 1926 | 83.1 |
| DeiT3-L | 224 | 304.4 | 59.7 | 535 | 84.8 |
| PoolFormer-M58 Yu et al. (2022) | 224 | 73.5 | 11.6 | 884 | 82.4 |
| Hybrid | | | | | |
| CoaT-Lite-S Xu et al. (2021a) | 224 | 19.8 | 4.1 | 2269 | 82.3 |
| CrossViT-B Chen et al. (2021a) | 240 | 105.0 | 20.1 | 1321 | 82.2 |
| Visformer-S Chen et al. (2021d) | 224 | 40.2 | 4.8 | 3676 | 82.1 |
| EdgeViT-S Pan et al. (2022) | 224 | 13.1 | 1.9 | 4254 | 81.0 |
| EfficientFormer-L7 Li et al. (2022) | 224 | 82.2 | 10.2 | 1359 | 83.4 |
| MaxViT-B Tu et al. (2022) | 224 | 120.0 | 23.4 | 507 | 84.9 |
| MaxViT-L Tu et al. (2022) | 224 | 212.0 | 43.9 | 376 | 85.1 |
| **FasterViT** | | | | | |
| **FasterViT-0** | 224 | 31.4 | 3.3 | **5802** | **82.1** |
| **FasterViT-1** | 224 | 53.4 | 5.3 | **4188** | **83.2** |
| **FasterViT-2** | 224 | 75.9 | 8.7 | **3161** | **84.2** |
| **FasterViT-3** | 224 | 159.5 | 18.2 | **1780** | **84.9** |
| **FasterViT-4** | 224 | 424.6 | 36.6 | **849** | **85.4** |
| **FasterViT-5** | 224 | 957.5 | 113.0 | **449** | **85.6** |
| **FasterViT-6** | 224 | 1360.0 | 142.0 | **352** | **85.8** |

uses it as key and value for local window attention. It has a complexity of $O(k^2 H^2 d + \frac{H^4}{k^2} d)$ (from the paper). Under the same size of the local window ($k$), and $H$, we can get the difference of $O(L + \frac{H^2 L^2}{k^4})$ for HAT and $O(\frac{H^2}{k^2})$ for Twins. HAT gets more efficient with higher resolution, for example, for $H = 32$, $k = 8$, with $L = 4$ we get $O(8)$ for HAT, whereas $O(16)$ for Twins.

## 4 RESULTS

### 4.1 IMAGE CLASSIFICATION

In Table 1, we demonstrate a quantitative comparison between the performance of FasterViT models and a variety of different hybrid, conv and Transformer-based networks on ImageNet-1K dataset. Comparing to Conv-based architectures, we achieve higher accuracy under the same throughput, for example, we outperform ConvNeXt-T by 2.2%. Considering the accuracy and throughput trade-off, FasterViT models are significantly faster than Transformer-based models such as the family of Swin

Table 2: Object detection and instance segmentation benchmarks using Cascade Mask R-CNN (He et al., 2017) on **MS COCO** dataset (Lin et al., 2014). All models employ $3\times$ schedule. All model statistics are reported using a input test resolution of $1280 \times 800$.

| Backbone | Throu. im/sec | $AP^{box}$ | | | $AP^{mask}$ | | |
|---|---|---|---|---|---|---|---|
| | | Box | 50 | 75 | Mask | 50 | 75 |
| Swin-T Liu et al. (2021) | 161 | 50.4 | 69.2 | 54.7 | 43.7 | 66.6 | 47.3 |
| ConvNeXt-T Liu et al. (2022b) | 166 | 50.4 | 69.1 | 54.8 | 43.7 | 66.5 | 47.3 |
| DeiT-Small/16 Touvron et al. (2021a) | 269 | 48.0 | 67.2 | 51.7 | 41.4 | 64.2 | 44.3 |
| **FasterViT-2** | **287** | **52.1** | **71.0** | **56.6** | **45.2** | **68.4** | **49.0** |
| Swin-S Liu et al. (2021) | 119 | 51.9 | 70.7 | 56.3 | 45.0 | 68.2 | 48.8 |
| X101-32 Xie et al. (2017) | 124 | 48.1 | 66.5 | 52.4 | 41.6 | 63.9 | 45.2 |
| ConvNeXt-S Liu et al. (2022b) | 128 | 51.9 | 70.8 | 56.5 | 45.0 | 68.4 | 49.1 |
| **FasterViT-3** | **159** | **52.4** | **71.1** | **56.7** | **45.4** | **68.7** | **49.3** |
| X101-64 Xie et al. (2017) | 86 | 48.3 | 66.4 | 52.3 | 41.7 | 64.0 | 45.1 |
| Swin-B Liu et al. (2021) | 90 | 51.9 | 70.5 | 56.4 | 45.0 | 68.1 | 48.9 |
| ConvNeXt-B Liu et al. (2022b) | 101 | 52.7 | 71.3 | 57.2 | 45.6 | 68.9 | 49.5 |
| **FasterViT-4** | **117** | **52.9** | **71.6** | **57.7** | **45.8** | **69.1** | **49.8** |

Transformers (Liu et al., 2021; 2022a). Furthermore, compared to hybrid models, such as the recent Efficient-Former (Li et al., 2022) and MaxViT (Tu et al., 2022) models, FasterViT on average has a higher throughput while achieving a better ImageNet top-1 performance. To validate the scalability of the proposed model, we pre-trained FasterViT-4 on ImageNet-21K dataset and fine-tuned it on various im-

| Model | Image Size (Px) | #Param (M) | FLOPs (G) | Throughput (Img/Sec) | Top-1 (%) |
|---|---|---|---|---|---|
| ViT-L/16$^{\ddagger}$ Liu et al. (2021) | 384 | 307.0 | 190.7 | 149 | 85.2 |
| Swin-L$^{\ddagger}$ Liu et al. (2021) | 224 | 197.0 | 34.5 | 787 | 86.3 |
| Swin-L$^{\ddagger}$ Liu et al. (2021) | 384 | 197.0 | 103.9 | 206 | 87.3 |
| ConvNeXt-L$^{\ddagger}$ Liu et al. (2022b) | 224 | 198.0 | 34.4 | 508 | 86.6 |
| ConvNeXt-L$^{\ddagger}$ Liu et al. (2022b) | 384 | 198.0 | 101.0 | 172 | 87.5 |
| **FasterViT-4$^{\ddagger}$** | 224 | 424.6 | 36.6 | **849** | **86.6** |
| **FasterViT-4$^{\ddagger}$** | 384 | 424.6 | 119.2 | **281** | **87.5** |

Table 3: **ImageNet-21K** pretrained classification benchmarks on **ImageNet-1K** dataset (Deng et al., 2009). Image throughput is measured on A100 GPUs with batch size of 128. $^{\ddagger}$ denotes models that are pre-trained on ImageNet-21K dataset.

age resolutions on ImageNet-1K dataset. As shown in Table 3, FasterViT-4 has a better accuracy-throughput trade-off compared to other counterparts.

## 4.2 DENSE PREDICTION TASKS

In Table 2, we present object detection and instance segmentation benchmarks on MS COCO dataset (Lin et al., 2014) with Cascade Mask R-CNN (He et al., 2017) network. We observe that FasterViT models have better accuracy-throughput trade-off when compared to other counterparts. Specifically, FasterViT-4 outperforms ConvNeXt-B and Swin-B by +0.2 and +1.0 in terms of box AP and +0.3 and +1.0 in terms of mask AP, while being 15% and 30% faster in terms of throughput, respectively. We also conduct additional object detection experiments with FasterViT-4

| Model | Throughput | FLOPs (G) | IoU(ss/ms) |
|---|---|---|---|
| Swin-T Liu et al. (2021) | 350 | 945 | 44.5/45.8 |
| ConvNeXt-T Liu et al. (2022b) | 363 | 939 | - /46.7 |
| **FasterViT-2** | **377** | 974 | **47.2/48.4** |
| Twins-SVT-B Chu et al. (2021a) | 204 | - | 47.7/48.9 |
| Swin-S Liu et al. (2021) | 219 | 1038 | 47.6/49.5 |
| ConvNeXt-S Liu et al. (2022b) | 234 | 1027 | - /49.6 |
| **FasterViT-3** | **254** | 1076 | **48.7/49.7** |
| Twins-SVT-L Chu et al. (2021a) | 164 | - | 48.8/50.2 |
| Swin-B Liu et al. (2021) | 172 | 1188 | 48.1/49.7 |
| ConvNeXt-B Liu et al. (2022b) | 189 | 1170 | - /49.9 |
| **FasterViT-4** | **202** | 1290 | **49.1/50.3** |

Table 4: Semantic segmentation on **ADE20K** (Zhou et al., 2017) with UPerNet (Xiao et al., 2018).

ImageNet-21K pre-trained backbone and the state-of-the-art DINO (Zhang et al., 2022) model and achieve a high detection accuracy of 58.7 box AP. In Table S.6, we present the semantic segmentation benchmarks with UPerNet (Xiao et al., 2018) network for experiments conducted on ADE20K dataset (Zhou et al., 2017). Similar to previous tasks, FasterViT models benefit from a better performance-throughput trade-off.

# 5 ABLATION

**EdgeViT and Twins** As shown in Table 5, we performed a comprehensive ablation study to validate the effectiveness of HAT by replacing all attention layers with attention mechanisms in EdgeViT (Pan et al., 2022) and Twins (Chu et al., 2021a) in the 3rd and 4th stages. For all model variants, Faster-ViT models with HAT achieve a better accuracy, sometimes by a significant margin. Twins achieves a higher throughput due to its small kernel size (*i.e.* $k = 2$), however, this significantly limits its accuracy. The better performance of HAT is attributed to

| Model | Attention | FLOPs (G) | Thr(Img/Sec) | Top-1 (%) |
|---|---|---|---|---|
| FasterViT-0 | Twins (Chu et al., 2021a) | 3.0 | 6896 | 80.8 |
| FasterViT-0 | EdgeViT (Pan et al., 2022) | 3.2 | 5928 | 81.0 |
| FasterViT-0 | **HAT** | 3.3 | 5802 | **82.1** |
| FasterViT-1 | Twins (Chu et al., 2021a) | 4.7 | 4949 | 82.1 |
| FasterViT-1 | EdgeViT (Pan et al., 2022) | 4.8 | 4188 | 82.5 |
| FasterViT-1 | **HAT** | 5.3 | 4344 | **83.2** |
| FasterViT-2 | Twins (Chu et al., 2021a) | 8.0 | 3668 | 82.9 |
| FasterViT-2 | EdgeViT (Pan et al., 2022) | 8.5 | 3127 | 83.4 |
| FasterViT-2 | **HAT** | 8.7 | 3161 | **84.2** |

Table 5: Ablation study on the effectiveness of HAT compared to EdgeViT (Pan et al., 2022) and Twins (Chu et al., 2021a) self-attention mechanisms. All attention blocks are replaced with the indicated attention type.

its learnable information aggregation/propagation via CTs, and direct access to dedicated CTs in windowed attention.

**Carrier Token Size** We investigated the effect of carrier token size and window size on the accuracy and image throughput of the model. We observed that increasing the carrier token size can improve the performance at the cost of decreased throughput, sometimes by a significant margin. In addition, increasing the window size slightly improves the Top-1 accuracy while also decreasing the throughput. In fact, increasing the window size does not scale properly to higher resolution

| Model | Pretrain W8, I256 acc | im/s | W12, I384 acc | im/s | W16, I512 acc | im/s | W24, I768 acc | im/s |
|---|---|---|---|---|---|---|---|---|
| SwinV2-T Liu et al. (2022a) | 81.8 | 1674 | 83.2 | 573 | 83.8 | 168 | 84.2 | 72 |
| SwinV2-S Liu et al. (2022a) | 83.7 | 633 | 84.8 | 338 | 85.4 | 153 | - | - |
| **FasterViT-2** | **84.3** | **2500** | **85.3** | **984** | **85.5** | **489** | **85.6** | **155** |
| SwinV2-B Liu et al. (2022a) | 84.2 | 499 | 85.1 | 251 | 85.6 | 115 | - | - |
| **FasterViT-4 256** | **85.3** | **653** | **86.0** | **254** | **86.1** | **133** | **86.0** | **44** |

Table 6: Quantitative comparison between higher resolution fine-tuning of FasterViT and SwinV2. FasterViT is more accurate on average by 0.9%, and faster by 2x.

images due to its significant impact on efficiency. As a result, HAT is a more effective and efficient mechanism that can be employed to model long-range spatial dependencies without sacrificing the throughput. Please refer to supplementary materials for more details.

**Plug-and-Play HAT** We employed HAT as a plug-and-play module with Swin-T model Table 7. This change results in +0.9 and +0.4% improvement in terms of mIoU and Top-1 accuracy on ImageNet classification and ADE20K segmentation tasks. In addition, improvements on MS COCO by +0.5 box AP and +0.6 mask AP on object detection and instance segmentation tasks, respectively. In addition, we also provide

| | ImageNet top-1 | Thr | COCO AP$^{box}$ | AP$^{mask}$ | Thr | ADE20k mIoU | Thr |
|---|---|---|---|---|---|---|---|
| Swin-T | 81.3 | 2758 | 50.4 | 43.7 | 161 | 44.5 | 350 |
| **Swin-T + HAT** | **81.7** | 2721 | **50.9** | **44.3** | 150 | **45.4** | 338 |

Table 7: Ablation study on the effectiveness of HAT as a plug-and-play module with Swin-T model for various CV tasks. Thr stands for throughput and is measure in image/sec.

throughput comparisons and show that HAT can be efficiently used with existing architectures with minimal overhead. Hence, it validates the effectiveness of HAT as a standalone self-attention.

# 6 CONCLUSION

In this work, we have presented a novel hybrid model, denoted as FasterViT, which achieves SOTA Pareto-front in terms of ImageNet Top-1 accuracy and throughput. We have extensively validated the effectiveness of FasterViT in downstream dense prediction tasks such as object detection, instance segmentation and semantic segmentation. Our benchmarks demonstrate better accuracy-throughput trade-off in comparison to counterpart models such as ConvNeXt and Swin Transformer.

## 7 ACKNOWLEDGEMENT

We thank Amanda Moran, Christopher Lamb, Sivakumar Thottakara, Sudeep Sabnis, Ranjitha Prasanna and other members of NVIDIA NGC team for providing highly-optimized GPU cloud infrastructures which were used for training and evaluation of FasterViT models.

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

## A  APPENDIX

## B  TRAINING SETTINGS

**Image Classification**   We employ the ImageNet-1K dataset (Deng et al., 2009) for classification that includes 1.2M and 50K training and validation images. The dataset has 1000 categories and we report the performance in terms of top-1 accuracy. In addition, we use ImageNet-21K dataset which has 14M images with 21841 classes for pretraining.

We train all FasterViT models by using LAMB optimizer (You et al., 2019) optimizer for 300 epochs with a learning rate of 5e-3 and a total batch size of 4096 using 32 A100 GPUs. For data augmentation, we follow same strategies as in previous efforts (Liu et al., 2022b; 2021). We also use Exponential Moving Average (EMA) which often improves the performance. Further details on training settings can be found in the appendix. For pre-training on ImageNet-21K, we train the models for 90 epochs with a learning rate of 4e-3. In addition, we fine-tune the models for 60 epochs with a learning rate of 7e-5.

**Detection and Segmentation**   We used the MS COCO dataset (Lin et al., 2014) to finetune a Cascade Mask-RCNN network (He et al., 2017) with pretrained FasterViT backbones.  For this purpose, we trained all models with AdamW (Loshchilov & Hutter, 2017) optimizer with an initial learning rate of 1e-4, a $3 \times$ schedule, weight decay of 5e-2 and a total batch size of 16 on 8 A100 GPUs.

**Semantic Segmentation**   For semantic segmentation, we employed ADE20K dataset (Zhou et al., 2017) to finetune an UperNet network (Xiao et al., 2018) with pre-trained FasterViT backbones. Specifically, we trained all models with Adam-W (Loshchilov & Hutter, 2017) optimizer and by using a learning rate of 6e-5, weight decay of 1e-2 and total batch size of 16 on 8 A100 GPUs.

## C  ROBUSTNESS ANALYSIS

In this section, we analyze the robustness of FasterViT models on different datasets. We test FasterViT model variants on ImageNet-A Hendrycks et al. (2021b), ImageNet-R Hendrycks et al. (2021a) and ImageNetV2 Recht et al. (2019) datasets. In addition, we did not perform any fine-tuning and simply employed the pre-trained ImageNet-1K Deng et al. (2009) weights for each model.  As shown in Table S.2, FasterViT demonstrates promising robustness performance on various datasets for each model variant. Specifically, FasterViT-3 outperforms comparable models such as ConvNeXt-B and Swin-B Liu et al. (2022b) by +7.5% and +8.4% on ImageNet-A Hendrycks et al. (2021b), +0.6% and +5.3% on ImageNet-R Hendrycks et al. (2021a) and +1.3% and +2.7% on ImageNetV2 Recht et al. (2019), respectively. For larger models, FasterViT-4 outperforms ConvNeXt-L Liu et al. (2022b) by +7.9%, +2.6% and +1.5% on ImageNet-A Hendrycks et al. (2021b), ImageNet-R Hendrycks et al. (2021a) and ImageNetV2 Recht et al. (2019), respectively, hence validating the effectiveness of the proposed model in various benchmarks. Similar trends can be observed for smaller models.

## D  ABLATION

### D.1  COMPONENT-WISE STUDY

Table S.1 shows per component ablation. Two settings are considered: (i) when the model is trained without the component, (ii) when the component is disabled after the model is trained. The first shows if the model can operate well without the component, while the second cases shows if the components is used in the final model.

We observe that changing the window resolution to $14 \times 14$ in the 3rd stage (effectively removing HAT by have a full global window) improves the model accuracy by $+0.1\%$ while scarifying 10% of throughput. Even though this setup shows better accuracy, it does not scale to high resolution, and HAT is required. Removing the HAT block from the architecture results in $-0.24\%$ accuracy drop for re-trained model and $-1.49\%$ for post training study at the benefit of 8% throughtput improvement. CT attention is another block of high importance, resulting in $-3.85\%$ post training

Table S.2: Robustness analysis of **ImageNet-1K** Deng et al. (2009) pretrained FasterViT models on ImageNet-A Hendrycks et al. (2021b), ImageNet-R Hendrycks et al. (2021a) and ImageNetV2 Recht et al. (2019) datasets.

| Model | Size (Px) | #Param (M) | FLOPs (G) | Throughput (Img/Sec) | Clean (%) | A (%) | R (%) | V2 (%) |
|---|---|---|---|---|---|---|---|---|
| **FasterViT-0** | 224 | 31.4 | 3.3 | **5802** | **82.1** | **23.9** | **45.9** | **70.9** |
| **FasterViT-1** | 224 | 53.4 | 5.3 | **4188** | **83.2** | **31.2** | **47.5** | **72.6** |
| Swin-T Liu et al. (2021) | 224 | 28.3 | 4.4 | 2758 | 81.3 | 21.6 | 41.3 | 69.7 |
| ConvNeXt-T Liu et al. (2022b) | 224 | 28.6 | 4.5 | 3196 | 82.0 | 24.2 | 47.2 | 71.0 |
| ConvNeXt-S Liu et al. (2022b) | 224 | 50.2 | 8.7 | 2008 | 83.1 | 31.3 | 49.5 | 72.4 |
| **FasterViT-2** | 224 | 75.9 | 8.7 | **3161** | **84.2** | **38.2** | **49.6** | **73.7** |
| Swin-S Liu et al. (2021) | 224 | 49.6 | 8.5 | 1720 | 83.2 | 32.5 | 44.7 | 72.1 |
| Swin-B Liu et al. (2021) | 224 | 87.8 | 15.4 | 1232 | 83.4 | 35.8 | 46.6 | 72.3 |
| ConvNeXt-B Liu et al. (2022b) | 224 | 88.6 | 15.4 | 1485 | 83.8 | 36.7 | 51.3 | 73.7 |
| **FasterViT-3** | 224 | 159.5 | 18.2 | **1780** | **84.9** | **44.2** | **51.9** | **75.0** |
| ConvNeXt-L Liu et al. (2022b) | 224 | 198.0 | 34.4 | 508 | 84.3 | 41.1 | 53.4 | 74.2 |
| **FasterViT-4** | 224 | 424.6 | 36.6 | **849** | **85.4** | **49.0** | **56.0** | **75.7** |
| **FasterViT-5** | 224 | 975.5 | 113.0 | **449** | **85.6** | **52.7** | **56.9** | **76.0** |
| **FasterViT-6** | 224 | 1360.0 | 142.0 | **352** | **85.8** | **53.7** | **57.1** | **76.1** |

removal. Attention bias is an important component of our system, resulting in $-0.31\%$ drop in the re-training scenario. Removing CT propagation, results in the requirement to pool and propagate features at every layer (similar to EdgeViT), that costs 7% of total inference and in lower accuracy $-0.16\%$. CT initialization is important to the network, as accuracy drops by $-0.48\%$ in post-training removal. Removing all components and having only CNN plus windowed vanilla transformer results in $-0.46\%$.

## D.2 SWINV2 COMPARISON

In the Table 6 we compare the performance of SwinV2 Liu et al. (2022a) and Faster-ViT models on large image resolution. The initial model is pretrained with an image resolution of $256^2$px for 300 epochs on ImageNet-1K. Then models are fine-tuned on a larger resolution (I) for an 30 epochs with various window sizes (W). Faster-ViT consistently demonstrates a higher image throughput, sometimes by a significant margin compared to Swin Transformer V2 model. Hence validating the effectiveness of the proposed hierarchical attention for high input resolution.

| Ablation | Trained from scratch | Post training removal | Throughput ratio |
|---|---|---|---|
| HAT block | -0.24% | -1.49% | 1.08 |
| CT attention | -0.13% | -3.85% | 1.00 |
| Attention Bias | -0.31% | -8.90% | 1.00 |
| CT propagation | -0.16% | - | 0.93 |
| 1D pos bias | -0.07% | -24.85% | 1.00 |
| CT initialization | -0.05% | -0.48% | 1.00 |
| Window 14×14 | +0.10% | - | 0.90 |

Table S.1: Ablation study on the effectiveness of different components of HAT.

## E ATTENTION MAPS

In Fig. S.1, we have illustrated the full attention maps of stage 3 layers for different FasterViT model variants. For this purpose, we use input images of size $224 \times 224 \times 3$ and ImageNet-1K Deng et al. (2009) trained FasterViT models. For each model, from the top to the bottom rows, we show the attention maps from the first to the final layer with an interval of a quarter of the total number of layers at stage 3 (*e.g.* layers 1, 4, 9 and 12 for FasterViT-4).

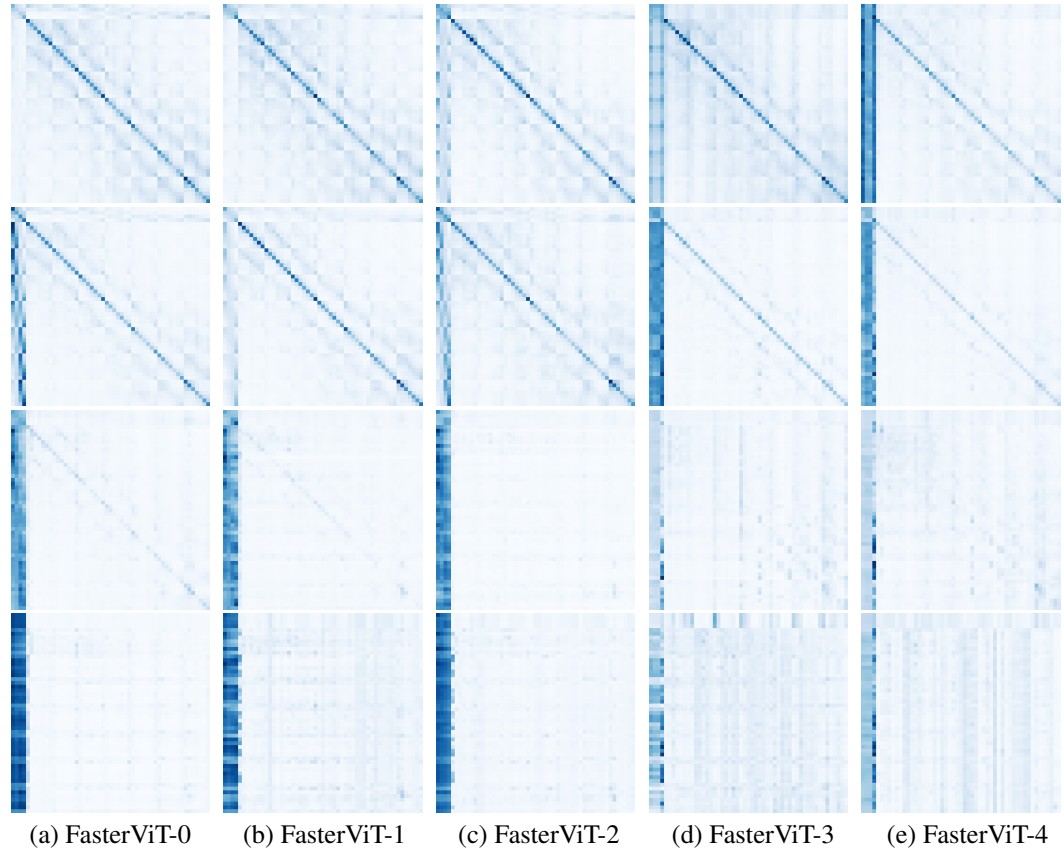

| (a) FasterViT-0 | (b) FasterViT-1 | (c) FasterViT-2 | (d) FasterViT-3 | (e) FasterViT-4 |

Figure S.1: (a) FasterViT-0. (b) FasterViT-1. (c) FasterViT-2. (d) FasterViT-3 (e) FasterViT-4. Full attention map visualizations of stage 3 for FasterViT model variants. From top to bottom, we visualize attention maps of first to last layers with an interval of a quarter length of the number of layers in stage 3 for each model. We visualize the attention maps of the same input image for all cases to facilitate comparability.

In particular, Stage 3 for this illustration serves an important purpose, since we use local attention windows of 7×7 with input features that have a resolution of 14 × 14. Hence, attention is computed in 4 local regions after window partitioning and 4 carrier tokens are designated to each corresponding window. Each illustrated attention map has a size of size 53×53 consisting of a concatenation of 4×4 carrier tokens and 49×49 local window-based attention. The carrier tokens are shown in in the top left position of each map. We observe that for all models, all tokens will attend to the carrier tokens with different patterns.

For FasterViT-0 and FasterViT-1 models, from the first to the last layers, all tokens transition to attend to the the carrier tokens (*i.e.* vertical bar on the left side). In the last layers, in addition to all tokens attending to the carrier tokens, we see a more global attention pattern, hence showing the cross interaction between different regions.

Figure S.2: Comparison of image throughput and ImageNet-1K Top-1 accuracy with TensorRT post-training model optimization. For all models, throughput is measured on A100 GPU with batch size of 1.

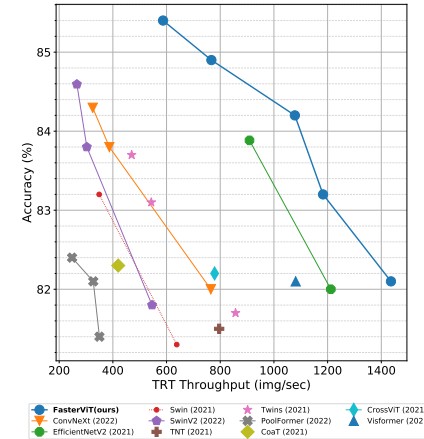

For FasterViT-2, FasterViT-3 and FasterViT-4 models, starting from the first layers, all tokens attend to both carrier and local tokens. In the last layers however, the attention pattern shifts from local to global. As discussed in this work and also shown in these illustrations, carrier tokens serve an integral role in modeling cross-region interactions and capturing long-range spatial dependencies.

## F  TENSORRT LATENCY

All throughput numbers and insights presented in the main paper were computed using PyTorch v1.13. In order to demonstrate the scalability with post-training optimization techniques,

we compared throughput using the TensoRT (TRT) framework for *batch size 1*, as illustrated in Fig S.2. FasterViT is still considerably faster than other models, making it a good choice to meet various efficient inference design targets.

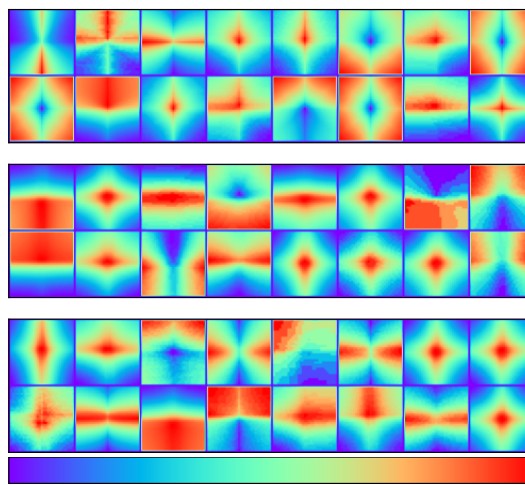

Figure S.3: Learned positional biases for attentions in the 3rd stage of FasterViT-4 model finetuned for 512×512 px. Each kernel corresponds to a bias for a single head in the multi-headed attention. Visualizations demonstrate that the model learns positional dependent features, while also sharing the pattern between pixels.

## G  ATTENTION BIAS

We follow the concept of relative positional bias in the attention from Swin Liu et al. (2021). Particularly, we use the implementation with MLP from SwinV2 Liu et al. (2022a), where relative coordinate shift in $x, y$ is transformed to the positional bias in the attention via 2-layer network. This allows the model to learn relative position aware kernels, and to introduce image inductive bias. We visualize learned positional

biases of the MLP in FasterViT-4 finetuned for 512 with window size of 16×16 pixels in Fig S.3. The visualization shows a diverse set of kernels learned by FasterViT model.

## H  FASTERVIT PROFILING

In Fig. S.4, we provide detailed stage-wise profiling of FasterViT-2 using NVIDIA DLSIM. As expected, stage 3 (HAT) has the highest latency, FLOPs and memory footprint since it is composed of considerably more layers compared to other stages.

## I  DESIGN INSIGHTS

*Layer normalization Ba et al. (2016).* We found it to be critical for transformer blocks (stage 3 and 4). Replacing it with batch normalization leads to accuracy drop of 0.7%. The LN performs cross token normalization and affects cross-channel interaction.

*No feature map reshaping.* In our architecture, we have removed windowing and de-windowing functions from transformer layers. They are usually used to perform convolutions between layers (like in Twins Chu et al. (2021a), EdgeViT Pan et al. (2022), Visformer Chen et al. (2021d)), or window shifting (Swin Liu et al. (2021), SwinV2 Liu et al. (2022a)). We perform windowing only once once in stages 3 and 4, and keep data as tokenized with channel last. This leads to throughput improvement of 5% for PyTorch and 10% for TensorRT.

*LAMB optimizer You et al. (2019).* We observed incredible stability of LAMB You et al. (2019) optimizer for training our biggest models (FasterViT-3 and FasterViT-4), more widely used AdamW Loshchilov & Hutter (2017) was leading to NaNs for some trainings. We attribute this to joined usage of batch normalization and layer normalization Ba et al. (2016) in the same model.

*Positional bias.* We employ 1D positional bias for local and carrier tokens, as well as 2D relative attention bias by MLP introduced in SwinV2 Liu et al. (2022a). For 1D bias we remove $log$ scale. This approach yields flexibility to the image size, as positional encoding is interpolated by MLP if resolution change. Those positional biases are quick to compute, however, will block all cores in GPUs until positional biases are computed, and will significantly impact the throughput. To address this, we propose to pre-compute positional biases for a given feature resolution and skip the MLP bottleneck, leading to 6% throughput gain.

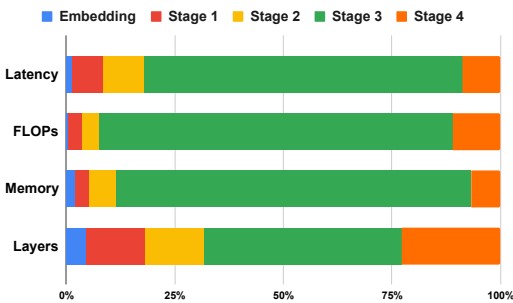

Figure S.4: **FasterViT-2** profiling benchmarks. Stage 3 (HAT) dominates over all metrics.

*Drop-out.* We found that conventional drop-out on MLP layers and attention has a negative effect on the final accuracy even for big models that overfit. Stochastic depth is helpful; in contrary to recent trends, we found that a small probability (up to 30%) works better than 65% like in DEIT3 Touvron et al. (2022). Better regularization can be achieved by increased weight decay. For example, model 4 with drop-path rate of 50% and weight decay of 0.05 achieves 84.91%, while model 4 with drop-path rate of 30% and weight decay of 0.12 achieves 85.15%.

*MESA Du et al..* It is shown to be useful to prevent overfitting of larger models at little overhead. MESA is a simplified version of SAM Foret et al. (2020) that forces optimization to have sharper minima at the convergence, naive implementation slows down training by 2x. In MESA, authors propose to simply apply knowledge distillation loss with respect to the EMA weight computed during training, the training overhead is almost not noticeable. We enable it after 25% of the training, coefficient is set proportionally to the model size in range 0.25 (FasterViT-0)-3.0(FasterViT-4).

*Intermediate LN.* SwinV2 Liu et al. (2022a) argues that intermediate LN Ba et al. (2016) help to stabilize training of large models, we saw accuracy degradation of this approach.

| | Output Size (Downs. Rate) | FasterViT-1 | FasterViT-2 | FasterViT-3 | FasterViT-4 |
|---|---|---|---|---|---|
| Stem | 112×112 (2×) | Conv-BN-ReLU C:32, S:2 × 1 | Conv-BN-ReLU C:64, S:2 × 1 | Conv-BN-ReLU C:64, S:2 × 1 | Conv-BN-ReLU C:64, S:2 × 1 |
| | | Conv-BN-ReLU C:80 × 1 | Conv-BN-ReLU C:96 × 1 | Conv-BN-ReLU C:128 × 1 | Conv-BN-ReLU C:196 × 1 |
| Stage 1 | 56×56 (4×) | LN-2D, Conv, C:160, S:2 | LN-2D, Conv, C:192, S:2 | LN-2D, Conv, C:256, S:2 | LN-2D, Conv, C:392, S:2 |
| | | ResBlock C:160 ×1, | ResBlock C:192 × 3, | ResBlock C:256 × 3, | ResBlock C:392 × 3, |
| Stage 2 | 28×28 (8×) | LN-2D, Conv, C:320, S:2 | LN-2D Conv, C:384, S:2 | LN-2D, Conv, C:512, S:2 | LN-2D, Conv, C:768, S:2 |
| | | ResBlock C:320 × 3, | ResBlock C:384 × 3, | ResBlock C:512 × 3, | ResBlock C:768 × 3, |
| Stage 3 | 14×14 (16×) | LN-2D, Conv, C:640, S:2 | LN-2D, Conv, C:768, S:2 | LN-2D, Conv, C:1024, S:2 | LN-2D, Conv, C:1568, S:2 |
| | | HAT C:640, head:8 × 8, | HAT C:768, head:8 × 8, | HAT C:1024, head:8 × 12, | HAT C:1568, head:16 × 12, |
| Stage 4 | 7×7 (32×) | LN-2D, Conv, C:1280, S:2 | LN-2D, Conv, C:1536, S:2 | LN-2D, Conv, C:2048, S:2 | LN-2D, Conv, C:3136, S:2 |
| | | HAT C:1280, head:16 × 5, | HAT C:1536, head:16 × 5, | HAT C:2048, head:16 × 5, | HAT C:3136, head:32 × 5, |

Table S.3: FasterViT architecture configurations. BN and LN-2D denote Batch Normalization and 2D Layer Normalization, respectively. HAT denotes Hierarchical Attention block.

## J    ARCHITECTURE DETAILS

In Table S.3, we show the different architecture configurations of the FasterViT model variants.

## K    CARRIER TOKEN SIZE

In Table S.4, we investigate the effect of carrier token size and window size on accuracy and latency of the model. We observe that increasing the carrier token window size can improve the performance at the cost of increased latency, sometimes by a significant margin. The 2x2 carrier token window size offers a great trade-off between accuracy and

Table S.5: **MS COCO** dataset (Lin et al., 2014) object detection results with DINO (Zhang et al., 2022) model. ‡ denotes models that are pre-trained on ImageNet-21K dataset.

| Backbone | Model | Epochs | FLOPs (G) | Throughput | AP$^{box}$ |
|---|---|---|---|---|---|
| Swin-L‡ (Liu et al., 2021) | HTC++ (Chen et al., 2019) | 72 | 1470 | - | 57.1 |
| Swin-L‡ (Liu et al., 2021) | DINO (Zhang et al., 2022) | 36 | 1285 | 71 | 58.5 |
| **FasterViT-4**‡ | DINO (Zhang et al., 2022) | 36 | 1364 | **84** | **58.7** |

latency. In addition, increasing the window size from 7 to 14 increases the Top-1 accuracy by +0.2%. However, as expected, it increases the latency by 10%. Hence, this shows the advantage of leveraging carrier token as an efficient mechanism to capture long-range contextual information. We also note that although increasing the window size results in better performance, it does not scale properly to higher resolution images. As a result, HAT is a more effective and efficient mechanism that can be employed without sacrificing image throughput.

| Window Size | Carrier Token Size | Latency Ratio | Top-1 (%) |
|---|---|---|---|
| 7 | 2 | 1 | 84.2 |
| 7 | 1 | 1.05 | 83.9 |
| 7 | 9 | 0.47 | 84.9 |
| 14 | 0 | 0.9 | 84.4 |

Table S.4: Effect of window and carrier token size on latency and Top-1 accuracy.

## L  DOWNSTREAM EXPERIMENTS

We provide additional experiments for both object detection and semantic segmentation with more models, across different sizes, to demonstrate the effectiveness and efficiency of our work. Firstly, in Table S.5, we present additional object detection experiments with DINO on MS-COCO dataset. The DINO model with FasterViT-4 is 18.30% faster than its counterpart with Swin-L backbone in terms of image throughput and outperforms it by +0.1 in terms of box AP.

We also added a semantic segmentation study on the ADE20K dataset with the FPN network, as shown below. Specifically, we compare against PoolFormer and PVT backbones. In this experiment, the model with FasterViT-1 backbone outperforms counterpart PoolFormer-S36 by +0.7 in terms of mIoU while also being 8.38% faster in terms of image throughput. Similarly, the model with FasterViT-2 backbone significantly outperforms PoolFomer-M36 counterpart by +1.1 in terms of mIOU while being 10.05% faster. We have added these experiments to the manuscript. We believe that the above experiments validate the effectiveness of FasterViT as an efficient backbone for downstream tasks such as segmentation and detection across different model sizes.

| Backbone | Model | Throughput | mIoU |
|---|---|---|---|
| PoolFomer-S36 (Yu et al., 2022) | FPN | 453 | 42.0 |
| **FasterViT-1** | FPN | **491** | **42.7** |
| PoolFomer-M36 (Yu et al., 2022) | FPN | 368 | 42.4 |
| **FasterViT-2** | FPN | **405** | **43.5** |

Table S.6: Semantic segmentation on **ADE20K** (Zhou et al., 2017) with FPN network.

## M  IMPACT OF CONV-BLOCKS ON THROUGHPUT

We conducted an additional ablation study to demonstrate the effect of conv-based block on both accuracy and throughput as shown below. According to our experiments, replacing Conv-based blocks with Transformer-based counterparts significantly reduces the throughput while also reducing the accuracy. As expected, the Conv-based blocks are more efficient than the transformer counterparts for processing larger input sizes. The model with conv-based blocks also has higher accuracy compared to their fully-transformer-based counterparts due to incorporating inductive biases such as locality. The combination of Conv-based (stage 1 and 2) and transformer-based (stage 3 and 4) architecture as presented in FasterViT strikes the right balance between accuracy and efficiency.

# N    THROUGHPUT ON DIFFERENT PLATFORMS

In order to validate the effectiveness of FasterViT on different platforms, we present additional throughput comparisons on different hardware such as NVIDIA V100, NVIDIA TITAN RTX and NVIDIA A6000 GPUs, Jetson Nano and Intel(R) Xeon(R) E5-2698 v4 CPU. For all comparisons, we use a batch size of 128, unless otherwise stated. Our benchmarks show that FasterViT achieves a Pareto-front for ImageNet Top-1 and throughput trade-off, hence validating the effectiveness and scalability of our model to different hardware platforms.

| Model | Top-1 | Throughput |
|---|---|---|
| FasterViT-0 | 82.1 | 5802 |
| FasterViT-0 wo Conv-block | 81.7 | 3616 |
| FasterViT-1 | 83.2 | 4188 |
| FasterViT-1 wo Conv-block | 82.8 | 3280 |
| FasterViT-2 | 84.2 | 3161 |
| FasterViT-2 wo Conv-block | 83.8 | 2085 |
| FasterViT-3 | 84.9 | 1780 |
| FasterViT-3 wo Conv-block | 84.5 | 1397 |
| FasterViT-4 | 85.4 | 849 |
| FasterViT-4 wo Conv-block | 84.9 | 712 |

Table S.7: Effect of Conv-based stages on throughput and accuracy of different FasterViT models.

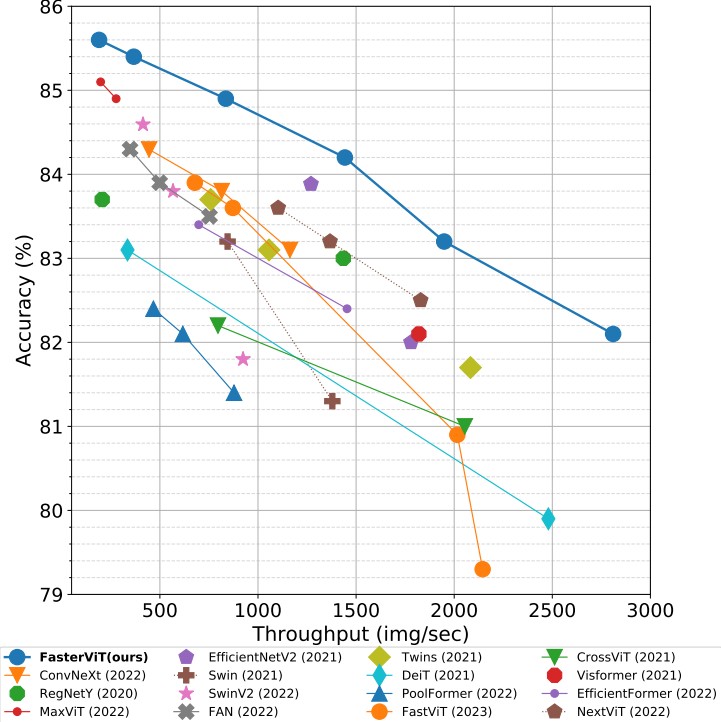

Figure S.5: Comparison of image throughput and ImageNet-1K Top-1 accuracy on NVIDIA V100 GPU for batch size of 128.

In Fig. S.5, we demonstrate the throughput and accuracy trade-off on V100 GPU and observe that FasterViT achieves a Pareto front. Additionally, in Fig. S.6, we illustrate the same comparison for NVIDIA TITAN RTX GPU, which is considered as an enthusiast-class graphics card. Surprisingly, we see that FasterViT attains a Pareto front on this platforms as well.

In addition, as shown in Fig. S.7, we report the throughput for all models using an NVIDIA A6000 GPU to confirm the scalability of our proposed architecture to various types of hardware. On A6000 GPU, FasterViT still demonstrates a strong performance and achieves a SOTA Pareto front except for an EfficientNetV2 variant which achieves a comparable performance to FasterViT-2.

In addition to GPU hardware, we have also measured throughput on a CPU device as well as NVIDIA Jetson Nano which is considered as an embedded system. In Fig. S.8, we demonstrate measurement for Top-1 and image throughput on Intel(R) Xeon(R) E5-2698 v4 CPU. On this device, we still

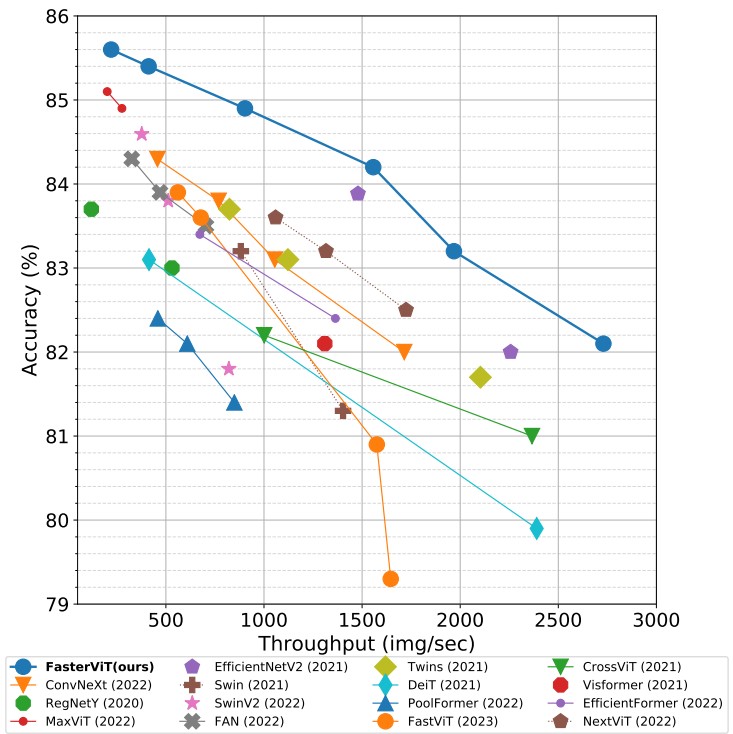

Figure S.6: Comparison of image throughput and ImageNet-1K Top-1 accuracy on NVIDIA TITAN RTX GPU for batch size of 128.

observe a dominant performance from different FasterViT variants. However, two variants from EfficientNetV2 and RegNetY models achieve a comparable performance to counterpart FasterViT models. In Fig. S.9, we present the throughput and accuracy tradeoff for NVIDIA Jetson Nano. Surprisingly, all FasterViT variants demonstrate a strong performance.

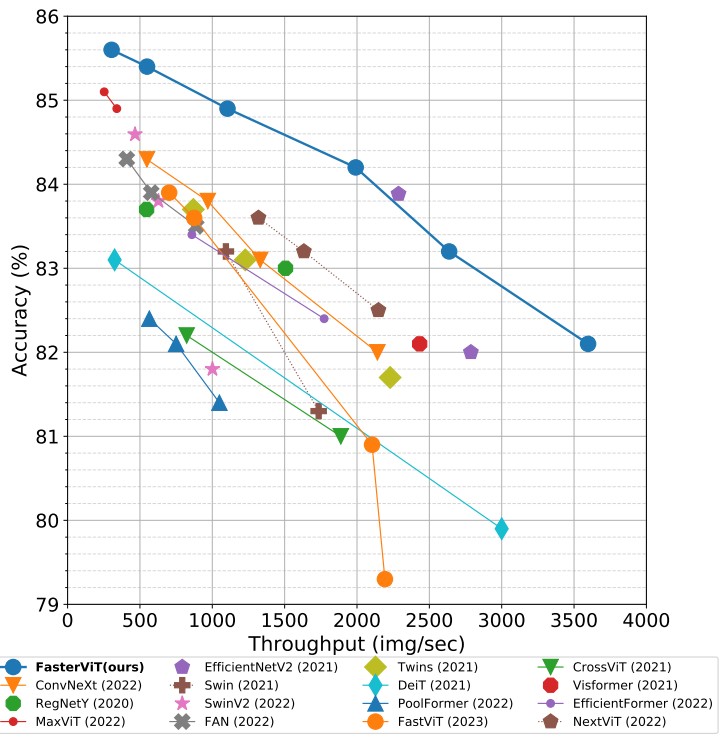

Figure S.7: Comparison of image throughput and ImageNet-1K Top-1 accuracy on NVIDIA A6000 GPU for batch size of 64.

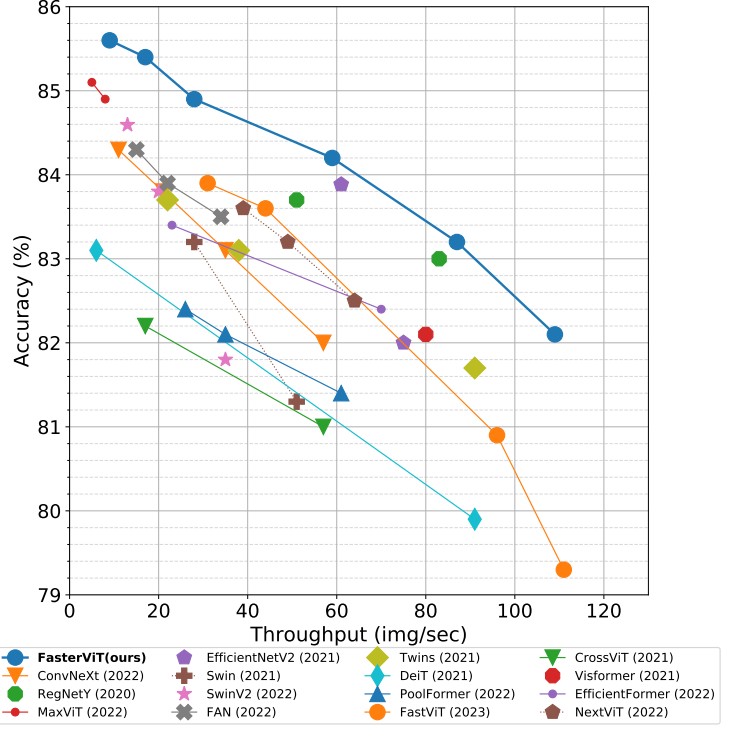

Figure S.8: Comparison of image throughput and ImageNet-1K Top-1 accuracy on Intel(R) Xeon(R) E5-2698 v4 CPU for batch size of 128.

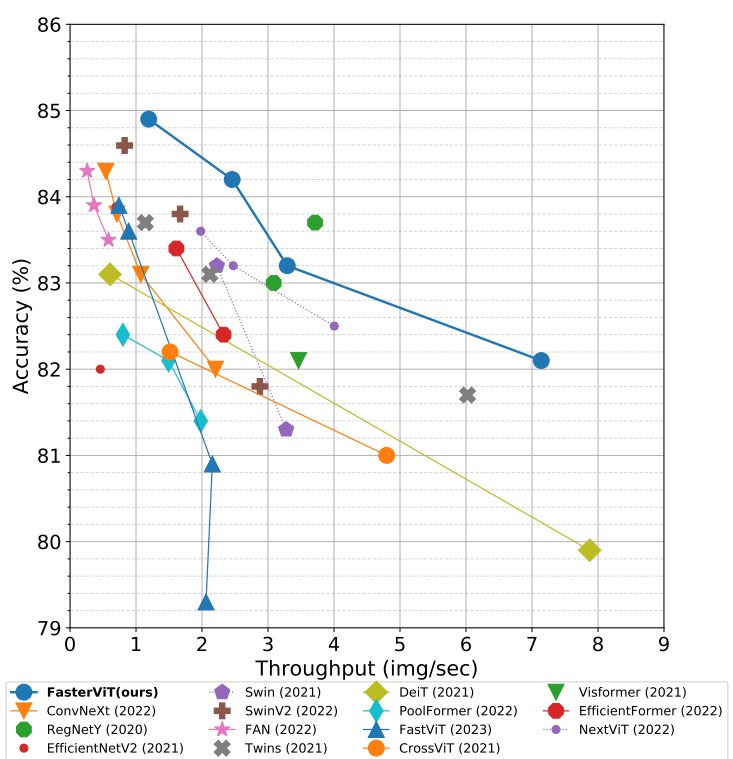

Figure S.9: Comparison of image throughput and ImageNet-1K Top-1 accuracy on NVIDIA Jetson Nano for batch size of 1.

