# OpenReview forum: "FasterViT: Fast Vision Transformers with Hierarchical Attention"
_ICLR.cc/2024/Conference — ICLR 2024 poster_

### Official Review · Reviewer_oF4Y · 2023-10-31

**Soundness:** 3 good
**Presentation:** 3 good
**Contribution:** 3 good
**Rating:** 6
**Confidence:** 4

**Summary:**

The paper introduces a new vision transformer architecture, FasterViT, aiming to improve the GPU throughput for vision tasks. The key innovations of the paper include i) using convolutional blocks for the early stages of the model, which are typically memory bound; ii) a new hierarchical attention module (HAT) that interleaves global attention over carrier tokens and self-attention within local windows. Experimental results show that, compared to prior works, FasterViT achieves a better trade-off between accuracy and GPU throughput.

**Strengths:**

(S1) [Method] the idea of incorporating convolutional blocks for memory-bound stages is interesting, which is actually similar to the design of MobileViT.

(S2) [Performance] Compared to the latest approaches, FasterViT presents improved throughputs on A100.

(S3) [Ablation] Ablations show that the proposed HAT is a plug-and-play module that can also improve the performance of other models

(S4) [Writing] The paper is easy to follow.

**Weaknesses:**

(W1) The proposed HAT looks similar to the hierarchical attention proposed in EdgeViT (Pan et al. 2022). While comparisons with EdgeViT is included, it would be beneficial if the author could further clarify the key difference between HAT and the attention block in EdgeViT (maybe just a small paragraph)

(W2) If the reviewer understood correctly, FasterViT incorporated two new designs to improve the throughout i) conv-blocks in memory-bound stages ii) HAT. Clarifying how each element contributes to the throughput would be insightful. Here, the impact of ii) is shown in Table 5, an ablation study for i) would be a valuable addition.

(W3) The authors are encouraged to expand the evaluation to include accelerators beyond A100, such as different GPUs, or even NPUs, so as to provide a broader performance perspective.


The reviewer appreciates the improved performance of FasterViT, but has some concerns about clarity and evaluation. At this moment, the reviewer would like to rate the paper as weak accept.

**Questions:**

N.A.

---

> ### Author Response · Authors · 2023-11-23
> **Response to Reviewer oF4Y**
>
> We sincerely thank the reviewer for their time and efforts in reviewing our work and providing valuable feedback that can further strengthen our manuscript. Below please find our detailed responses:
>
> > **Differences between HAT and attention block in EdgeViT**
>
> There are key differences between our proposed HAT and the attention block in EdgeViT. The attention in EdgeViT follows a “local-global-local” paradigm. Specifically, a convolutional layer first aggregates local information into representative tokens. A global (sparse) attention is then computed on the aggregated tokens. Lastly, a depthwise convolutional layer is used to compute local information from global information. In contrast, HAT computes self-attention between: (1) each carrier token and their corresponding local regions (2) all carrier tokens. The carrier tokens are only computed once and propagated to next layers, whereas EdgeViT repeats the computation for each operation at every transformer block.  One disadvantage of the proposed scheme in EdgeViT is the loss of information in transitioning from one attention block to another, which is sub-optimal. Another potential drawback is that the way information is propagated is restricted by convolution operations; meanwhile, HAT allows one to learn the aggregation and propagation mechanics via attention.
>
> > **Impact of conv-blocks on throughput**
>
> Thank you for your comment. We have conducted an additional ablation study to demonstrate the effect of conv-based block on both accuracy and throughput as shown below.
>
> | Model   |Top-1 | Throughput
> |-------------------|----------------|----------------|
> | FasterViT-0 |82.1|5802|
> | FasterViT-0 wo Conv-block |81.7|3616|
> | FasterViT-1|83.2|4188|
> | FasterViT-1 wo Conv-block |82.8|3280|
> | FasterViT-2 |84.2|3161|
> | FasterViT-2 wo Conv-block |83.8|2085|
> | FasterViT-3 |84.9|1780|
> | FasterViT-3 wo Conv-block |84.5|1397|
> | FasterViT-4 |85.4|849|
> | FasterViT-4 wo Conv-block |84.9|712|
>
>
> According to our experiments, replacing Conv-based blocks with Transformer-based counterparts significantly reduces the throughput while also reducing the accuracy. As expected, the Conv-based blocks are more efficient than the transformer counterparts for processing larger input sizes. The model with conv-based blocks also has higher accuracy compared to their fully-transformer-based counterparts due to incorporating inductive biases such as locality.
>
>
> The combination of Conv-based (stage 1 and 2) and transformer-based (stage 3 and 4) architecture as presented in FasterViT strikes the right balance between accuracy and efficiency.
>
> We have added this ablation study to the revised manuscript.
>
>
> > **Throughput comparison on other platforms**
>
> We have added additional throughput comparison on different platforms such as V100, TITAN RTX and A6000 GPUs, Jetson Nano
> and Intel(R) Xeon(R) E5-2698 v4 CPU. For all comparisons, we show that FasterViT achieves a Pareto-front in for ImageNet Top-1 and throughput trade-off, hence validating the effectiveness and scalability of our model to different hardware platforms.
>
>
> In V100 ([Figure](https://drive.google.com/file/d/1FP0pDnZMdxw2Sy0z1tX-8A54T6eKD0cq/view?usp=sharing)) or TITAN RTX ([Figure](https://drive.google.com/file/d/1elnLkDMM5s2ch-E5MKb5YDj1vV_Rsx1s/view?usp=sharing)) comparisons for instance, we observe a strong performance for all FasterViT models. Even on CPU ([Figure](https://drive.google.com/file/d/1iVsev1XxgNaFEgFrhV0i4lGO-Fldl3z9/view?usp=sharing)), FasterViT still outperforms competitors by a significant margin.
>
> Please see the supplementary materials of the revised manuscript for all the figures regarding throughput and Top-1 accuracy.

---

### Official Review · Reviewer_Wvdn · 2023-11-01

**Soundness:** 3 good
**Presentation:** 3 good
**Contribution:** 3 good
**Rating:** 6
**Confidence:** 4

**Summary:**

The paper introduces a novel family of hybrid CNN-ViT neural networks called FasterViT, designed for high image throughput in computer vision applications. FasterViT leverages the strengths of both CNNs and ViT by introducing a Hierarchical Attention (HAT) approach that decomposes global self-attention into a multi-level attention with reduced computational costs. This approach efficiently combines local and global representation learning. FasterViT consists of four stages, reducing input image resolution while doubling feature maps in the early stages and employing transformer blocks in later stages. The proposed HAT mechanism efficiently captures long-range spatial dependencies and cross-window interactions. Extensive validation on various computer vision tasks, including image classification, object detection, and segmentation, demonstrates that FasterViT achieves state-of-the-art performance in terms of accuracy and image throughput, especially for high-resolution images, outperforming competitive models like Swin Transformer V2.

**Strengths:**

- FasterViT is tailored for high-resolution input images and demonstrates faster image throughput compared to competitive models, particularly in handling images with higher resolutions.
- The proposed HAT approach efficiently decomposes global self-attention into a multi-level attention mechanism, reducing computational complexity and enabling effective local and global representation learning. Overall, the idea of carrier tokens is novel and interesting.
- The paper extensively validates FasterViT on various computer vision tasks, including image classification, object detection, and semantic segmentation, showcasing its state-of-the-art performance across a wide range of applications.

**Weaknesses:**

- The comparisons in Table 5 and 7 demonstrate minor improvements in terms of accuracy, while the throughput in Table 5 is reduced when using HAT.
- Performance is compared on A100 GPUs. More platforms should be used to see if throughput results are consistent. At present results are not conclusive.

**Questions:**

- The authors should better explain the difference between the proposed attention and window local approaches line Swin.

---

> ### Author Response · Authors · 2023-11-23
> **Response to Reviewer Wvdn**
>
> We sincerely thank the reviewer for their time and efforts in reviewing our work and providing valuable feedback that can further strengthen our manuscript. Below please find our detailed responses:
>
> > **Differences between proposed attention and local approaches like Swin**
>
> Specifically, Swin uses a local-only window-based attention and window shifting. However, window shifting is sub-optimal for modeling cross-region interactions due to the limited stride. On the contrary, in our work, we use carrier tokens to model cross-region interactions (global attention) as well as interaction between the carrier token and local regions. Hence, this allows for effectively modeling both short and long range dependencies in a very efficient manner.
>
> > **Minor improvements in terms of accuracy while Reduced throughput when using HAT**
>
> Thank you for your comment. We believe that accuracy improvements in Table 5 are significant when compared to other models. Specifically, for FasterViT-1, using HAT improves the performance by 1.1% in terms of Top-1 accuracy. For comparison, a mere 1.1% improvement in terms of ImageNet Top-1 accuracy equates to scaling from ConvNeXt-T (82.1%) to ConvNeXt-S (83.2%) models. Furthermore, the drop in the throughput is expected since HAT effectively models the cross-region interactions that may not be properly captured by Twins and EdgeViT.
>
> > **Throughput comparison on other platforms**
>
> We have added additional throughput comparison on different platforms such as V100, TITAN RTX and A6000 GPUs, Jetson Nano
> and Intel(R) Xeon(R) E5-2698 v4 CPU. For all comparisons, we show that FasterViT achieves a Pareto-front in for ImageNet Top-1 and throughput trade-off, hence validating the effectiveness and scalability of our model to different hardware platforms.
>
>
> In V100 ([Figure](https://drive.google.com/file/d/1FP0pDnZMdxw2Sy0z1tX-8A54T6eKD0cq/view?usp=sharing)) or TITAN RTX ([Figure](https://drive.google.com/file/d/1elnLkDMM5s2ch-E5MKb5YDj1vV_Rsx1s/view?usp=sharing)) comparisons for instance, we observe a strong performance for all FasterViT models. Even on CPU ([Figure](https://drive.google.com/file/d/1iVsev1XxgNaFEgFrhV0i4lGO-Fldl3z9/view?usp=sharing)), FasterViT still outperforms competitors by a significant margin.
>
> Please see the supplementary materials of the revised manuscript for all the figures regarding throughput and Top-1 accuracy.

---

### Official Review · Reviewer_tsUS · 2023-11-02

**Soundness:** 3 good
**Presentation:** 3 good
**Contribution:** 3 good
**Rating:** 5
**Confidence:** 5

**Summary:**

This paper proposes a fast ViT architecture using hierarchical attention module (HAT). The HAT module is a modified version of window attention by introducing a new carrier token that summarizes the information of a local window. In this way, the model can preserve a certain level of global information while being more efficient than conventional global attention. With such architectural changes, the proposed FasterViT is able to achieve fast inference on modern GPUs and produces good accuracy on classification and downstreaming tasks.

**Strengths:**

- The paper is overall well written and organized. The motivation of proposing the HTA module is technically sound - it is a good practice to do more memory intensive operations in early stages while putting the computational intensive operations to the later stage. This is also verified in the experiments, where the proposed model is more GPU friendly compared to existing models.
- The HAT module is not unnecessarily complex and intuitively easy to implement. It is also shown that it can be a plug-in replacement for conventional attention blocks, which makes it more flexible.

**Weaknesses:**

The major weakness of this work is experiment. The paper claims that with HAT and all the optimization regarding model architecture, the model has much less complexity compared to conventional attention and it compares the proposed model to a few efficient ViT. However, the comparison is on A100 GPU only and there is no comparison on any other platforms. Some recent works, such as EfficientFormer, FastViT and NextViT compared in Figure 1, all benchmarked their models on different mobile platforms. Therefore, it is unclear whether the proposed change applies similarly to other platforms, which could be very different from A100 GPu due to various software and hardware optimizations. In addition, the ablation study is not sufficient. To really understand the HAT module, a more detailed analysis on it should be provided.

**Questions:**

- While the proposed HAT is claimed to be more efficient, only benchmarks on A100 GPUs using a batch 128 are provided. Due to various software and hardware differences, conclusion drawn from the current benchmark may not generalize. For example, it is well known that depthwise conv is more efficient on mobile CPU than on GPU and some ops such as reshape/transpose on CPU is slower due to L2 cache limitation. Even on GPU only, the performance may also vary on different variants. Given that most recent works on efficient ViT have presented benchmark results on multiple platforms - GPU, CPU and DSP, the evaluation from this work is worse and incomplete. I would like to see more benchmarks otherwise the current results are not so convincing.
- The comparison with existing models is not complete. Although in Figure 1, the proposed model is compared to many recent works in terms of image classification task, the comparison on other downstreaming tasks such as detection and segmentation is quite limited. For example, it is only compared with 3 other models (which are not even designed for efficiency) on the semantic segmentation task.
- The ablation study only shows that the HAT module can be plugged into other model architecture and improves the performance marginally, but misses analysis on the module itself: how do you determine the architecture for the model variants from 0 to 6? Are there any guidelines or empirical results? How does the latency change with respect to changing the parameters in HAT, such as window size, conv kernel size, pooling size, etc? The current setting seems quite ad-hoc.
- The parameter count and flops of the proposed model seems on the higher end. Under similar latency, it's much larger than most models. This may prevent it from being adopted in realistic scenarios, where there may be restrictions on model size and memory footprint. How do you aim to resolve this?

---

> ### Author Response · Authors · 2023-11-23
> **Response to Reviewer tsUS**
>
> We sincerely thank the reviewer for their time and efforts in reviewing our work and providing valuable feedback that can further strengthen our manuscript. Below please find our detailed responses:
>
> > **Throughput comparison on other platforms**
>
> We have added additional throughput comparison on different platforms such as V100, TITAN RTX and A6000 GPUs, Jetson Nano
> and Intel(R) Xeon(R) E5-2698 v4 CPU. For all comparisons, we show that FasterViT achieves a Pareto-front in for ImageNet Top-1 and throughput trade-off, hence validating the effectiveness and scalability of our model to different hardware platforms.
>
>
> In V100 ([Figure](https://drive.google.com/file/d/1FP0pDnZMdxw2Sy0z1tX-8A54T6eKD0cq/view?usp=sharing)) or TITAN RTX ([Figure](https://drive.google.com/file/d/1elnLkDMM5s2ch-E5MKb5YDj1vV_Rsx1s/view?usp=sharing)) comparisons for instance, we observe a strong performance for all FasterViT models. Even on CPU ([Figure](https://drive.google.com/file/d/1iVsev1XxgNaFEgFrhV0i4lGO-Fldl3z9/view?usp=sharing)), FasterViT still outperforms competitors by a significant margin.
>
> Please see the supplementary materials of the revised manuscript for all the figures regarding throughput and Top-1 accuracy.
>
>
>
> > **Additional Downstream Benchmarks**
>
> We thank the reviewer for their comment. We provide additional experiments for both object detection and semantic segmentation with more models, across different sizes, to demonstrate the effectiveness and efficiency of our work.
>
> Firstly, we present additional object detection experiments with DINO on MS-COCO dataset as shown below:
>
> | Backbone   |Model|AP_box |Throughput
> |-------------------|----------------|---------------|---------------|
> |Swin-L |HTC++ |57.1|-|
> |Swin-L |DINO|58.5|71|
> |**FasterViT-4**|DINO|58.7|84|
>
> The DINO model with FasterViT-4 is 18.30% faster than its counterpart with Swin-L backbone in terms of image throughput and outperforms it by +0.1 in terms of box AP.
>
> We also added a semantic segmentation study on the ADE20K dataset with the FPN network, as shown below. Specifically, we compare against PoolFormer and PVT backbones:
>
>
> | Backbone   |Model|mIoU |Throughput
> |-------------------|----------------|---------------|---------------|
> |PoolFomer-S36 |FPN |42.0|453|
> |**FasterViT-1**|FPN|42.7|491|
> |PoolFomer-M36 |FPN |42.4|368|
> |**FasterViT-2**|FPN|43.5|405|
>
>
> The model with FasterViT-1 backbone outperforms counterpart PoolFormer-S36 by +0.7 in terms of mIoU while also being 8.38% faster in terms of image throughput. Similarly, the model with FasterViT-2 backbone significantly outperforms PoolFomer-M36 counterpart by +1.1 in terms of mIOU while being 10.05% faster.
>
>
> We believe that the above experiments validate the effectiveness of FasterViT as an efficient backbone for downstream tasks such as segmentation and detection across different model sizes.
>
>
> > **How does latency change with respect to change of HAT parameters**
>
> Thank you for your question. We have conducted an additional ablation study to investigate the effect of HAT hyperparameters on the latency and accuracy of the model. Specifically, we examine the effect of window size (W) and carrier token window size (W_C) as shown below for 224x224 resolution. We use FasterViT-2 with 7x7 window size and 2x2 carrier token window size as the base model. Latency ratio is measured against this base model.
>
>
> | Resolution   |Window Size|Carrier Token Window Size |Latency Ratio|Top1|
> |-------------------|----------------|---------------|---------------|---------------|
> |224 |7 |2|1|84.2|
> |224 |7 |1|1.05|83.9|
> |224 |7 |5|0.67|84.3|
> |224 |7 |6|0.57|84.4|
> |224 |7 |9|0.47|84.9|
> |224 |14 |0|0.9|84.4|
>
> We observe that increasing the carrier token window size can improve the performance at the cost of increased latency, sometimes by a significant margin. The 2x2 carrier token window size offers a great trade-off between accuracy and latency.
>
> In addition, increasing the window size from 7 to 14 increases the Top-1 accuracy by +0.2%. However, as expected, it increases the latency by 10%. We also note that although increasing the window size results in better performance, it does not scale properly to higher resolution images. As a result, HAT is a more effective and efficient mechanism that can be employed without sacrificing image throughput.
>
> > **How do you determine the architecture for the model variants from 0 to 6**
>
> Key design choices for each model variant are : (1) stage-wise depth (2) embedding dimension (3) number of attention heads. Our base model starts with a simple configuration that consists of 2, 3, 6, 5 layers per stage 1, 2, 3 and 4 respectively, an embedding dimension of 64 and 2, 4, 8, 16 attention heads for stages 1, 2, 3 and 4. From this base configuration, we increased the model capacity by increasing the embedding dimension or stage-wise depth to create a Pareto front of Top-1 and throughput trade-off.

---

### Official Review · Reviewer_B19F · 2023-11-02

**Soundness:** 3 good
**Presentation:** 3 good
**Contribution:** 3 good
**Rating:** 6
**Confidence:** 4

**Summary:**

This paper proposes a hybrid conv+transformer network, optimizing throughputs for high-resolution image processing.  It proposes a Hierarchical Attention to combine local window tokens and carrier tokens (one per window).  Results are mostly on ImageNet and ImageNet21k, and the results in Figure 1 are quite impressive.

**Strengths:**

1. Intuitive idea:  combing conv + attention is not new, and is often considered more efficient than pure conv or pure transformer for image processing. Window attention is also not new, but the hierarchical window attention is interesting.
2. Impressive results:  results in Figure 1 are pretty impressive. FastViT outperforms other models by pretty good margin on ImageNet.
3. Well written paper and easy to follow.

**Weaknesses:**

1. It is unclear how significant the proposed hierarchical attention (HAT) is.  Table 5 shows this HAT is better than Twins and EdgeViT; however, Table 7 show only marginal gains when comparing to the vanilla SwinTransformer if treating HAT as a plug-and-play module.
2. The idea of HAT is not well motivated.  Though it shows good empirical results, it is unclear why simply adding a per-window CT token can significantly improve quality.  Would be nice to add a few more ablation studies or insights.
3. All latencies are measured on A100 GPU. It would be nice to measure the latency on more diverse hardware platforms, such as different kinds of GPUs, and CPUs.
4. Some image texts (e.g. in Figure 4 and 5) are too small to read. I recommend enlarging the text in these images.

**Questions:**

1. Could you provide more intuition and insights why simply adding one CT token per window will significantly improve model quality?
2. Could you show the comparisons of FastViT with and without hierarchical attention for ImageNet, e.g., one is vanilla SwinTransformer-style attention and one is HAT?
3. In Table 7, could you also add the throughput comparison?  I am curious about the overhead of HAT.
4. Could you add latency for more hardwares (different GPUs and CPUs)?

---

> ### Author Response · Authors · 2023-11-23
> **Response to Reviewer B19F**
>
> We sincerely thank the reviewer for their time and efforts in reviewing our work and providing valuable feedback that can further strengthen our manuscript. Below please find our detailed responses:
>
> > **Intuition on why HAT and CT improve the model quality**
>
> HAT allows for modeling global contextual information with minimal impact on computational efficiency. Specifically, the carrier tokens in HAT facilitate capturing cross-region interactions in an efficient and scalable manner, especially for high-resolution images. Without carrier tokens, the self-attention can only process local information inside every window and lack the capability of capturing long-range information.
>
> > **Comparison of FasterViT with and without hierarchical attention for ImageNet**
>
> We have conducted an additional ablation study to demonstrate the effect of removing HAT for every FasterViT variant as shown below:
>
> | Model   |HAT |Top-1 |
> |-------------------|----------------|----------------|
> | FasterViT-0 |Yes|82.1|
> | **FasterViT-0 wo HAT** |No|81.7|
> | FasterViT-1 |Yes|83.2|
> | **FasterViT-1 wo HAT** |No|82.8|
> | FasterViT-2 |Yes|84.2|
> | **FasterViT-2 wo HAT** |No|83.7|
> | FasterViT-3 |Yes|84.9|
> | **FasterViT-3 wo HAT** |No|84.2|
> | FasterViT-4 |Yes|85.4|
> | **FasterViT-4 wo HAT** |No|84.3|
>
>
> We observe that removing HAT can significantly decrease the accuracy, especially for larger variants such as the FasterViT-4 model.
>
>
> In addition to the above experiments, Figure S.1 offers more insights on the effectiveness of HAT. Specifically, in this figure, the carrier tokens have been appended to the left top corner. The vertical line in  dense attention maps indicates that all other tokens append to the carrier tokens, hence allowing for cross-region communications to capture both short and long-range spatial dependencies.
>
>
> > **Significance of HAT and plug-and-play module gains in Swin Transformers**
>
>  Swin Transformers use window shifting to capture the local cross-region interactions. However, this scheme is sub-optimal as it is limited to a small area of coverage which cannot effectively capture the long-range spatial dependencies. However, HAT proposes an efficient and effective way of modeling global information which is important for downstream tasks and high-resolution images.
> The improvements shown in Table 7 are significant as no additional hyper-parameter tuning was done. Specifically, as shown in In Table 7, using HAT with Swin improves the performance by +0.9 in terms of mIoU for semantic segmentation. Similarly, it improves the performance by +0.5 and +0.6 interim of box AP and mask AP for object detection and instance segmentation. In addition, FasterViT-4 outperforms counterpart Swin-L model by +0.2% when fine tuning on 384x384 resolution.
>
> > **Throughput numbers and HAT overhead in Table 7**
>
> We provide throughput numbers for experiments in Table 7 in the following:
>
> | Model   |Top-1 |Throughput (ImageNet) |AP_box|AP_mask|Throughput (det)|mIoU|Throughput (seg)|
> |-------------------|----------------|----------------|----------------|----------------|----------------|----------------|----------------|
> | Swin-T |81.3|2758|50.4|43.7|161|44.5|330|
> | **Swin-T+HAT** |81.7|2721|50.9|44.3|150|45.4|338|
>
> As seen above, HAT has a minor impact on overhead across different tasks and can be used as a viable standalone self-attention mechanism.
>
> We have also added throughput numbers in Table 7 of the revised manuscript (all numbers are shown in red).
>
> > **Throughput comparison on other platforms**
>
> We have added additional throughput comparison on different platforms such as V100, TITAN RTX and A6000 GPUs, Jetson Nano
> and Intel(R) Xeon(R) E5-2698 v4 CPU. For all comparisons, we show that FasterViT achieves a Pareto-front in for ImageNet Top-1 and throughput trade-off, hence validating the effectiveness and scalability of our model to different hardware platforms.
>
>
> In V100 ([Figure](https://drive.google.com/file/d/1FP0pDnZMdxw2Sy0z1tX-8A54T6eKD0cq/view?usp=sharing)) or TITAN RTX ([Figure](https://drive.google.com/file/d/1elnLkDMM5s2ch-E5MKb5YDj1vV_Rsx1s/view?usp=sharing)) comparisons for instance, we observe a strong performance for all FasterViT models. Even on CPU ([Figure](https://drive.google.com/file/d/1iVsev1XxgNaFEgFrhV0i4lGO-Fldl3z9/view?usp=sharing)), FasterViT still outperforms competitors by a significant margin.
>
> Please see the supplementary materials of the revised manuscript for all the figures regarding throughput and Top-1 accuracy.
>
>
> > **Small text in figures**
>
> Thank you for your suggestion. We have improved the figures and enlarged their embedded text.

---

### Author Response · Authors · 2023-11-23
**Global Response**

We are grateful to the reviewers for their thoughtful and constructive feedback. We are delighted that they found the paper to be well-written and easy to follow and that they appreciated the novelty of the proposed self-attention module, the importance of the contributions in incorporating HAT into the FasterViT network, and the completeness of the experiments with strong SOTA performance. In addition to this global response, we have responded to each reviewer’s comments separately and have made the following changes to our manuscript per reviewer’s suggestions:


* Comprehensive throughput comparisons on different platforms such as V100, TITAN RTX and A6000 GPUs, Jetson Nano and Intel(R) Xeon(R) E5-2698 v4 CPU.
* Additional throughput numbers in Table 7 for classification, detection and segmentation experiments for Swin and Swin + HAT models.
* Additional downstream benchmarks for semantic segmentation and object detection.
* Ablation study to investigate the effect of window and carrier token size on throughput and accuracy.
* Ablation study to investigate the effect of conv-based stages on throughput and accuracy.


## Contributions of our work


In this work, we present a novel model, denoted as FasterViT, which is designed specifically for efficiency vs throughput trade-off on GPU hardware. We also introduce a novel Hierarchical Attention mechanism . We achieve significant improvements compared to other models. We have also demonstrated the scalability of FasterViT for various platforms such as CPU, Jetson Nano, A100, V100, A6000 and TITAN RTX GPUs.

We have already made our code available to the reviewers in the following:

https://bit.ly/FasterViT

---

### Meta-Review · Area_Chair_s9q4 · 2023-12-05

**Metareview:**

This submission received three positive scores and one negative score. After reading the paper, the review comments and the rebuttal, the AC thinks the major concerns about experiments have been sufficiently addressed by authors by rebuttal. Particularly, they provided comparisons for downstream tasks under various models sizes, and additional ablation studies to analyze the effect of hyper-parameters (e.g. window size) on the ImageNet Top-1 classification accuracy. The novelty of the proposed approach is conformed by reviewers.

**Justification For Why Not Higher Score:**

N/A

**Justification For Why Not Lower Score:**

N/A

---

### Decision · Program_Chairs · 2024-01-16

Accept (poster)